# The deubiquitylase OTUD3 stabilizes GRP78 and promotes lung tumorigenesis

Tongde Du[1,6], Hongchang Li[1,6], Yongsheng Fan[2], Lin Yuan[3], Xiaodan Guo[2], Qiong Zhu[1,2], Yuying Yao[1,2], Xin Li[4], Chunlei Liu[4], Xinhe Yu[5], Zhaofei Liu [5], Chun-Ping Cui[1], Chuanchun Han[2] & Lingqiang Zhang[1]

The deubiquitylase OTUD3 plays a suppressive role in breast tumorigenesis through stabilizing PTEN protein, but its role in lung cancer remains unclear. Here, we demonstrate that in vivo deletion of OTUD3 indeed promotes breast cancer development in mice, but by contrast, it slows down Kras[G12D]-driven lung adenocarcinoma (ADC) initiation and progression and markedly increases survival in mice. Moreover, OTUD3 is highly expressed in human lung cancer tissues and its higher expression correlates with poorer survival of patients. Further mechanistic studies reveal that OTUD3 interacts with, deubiquitylates and stabilizes the glucose-regulated protein GRP78. Knockdown of OTUD3 results in a decrease in the level of GRP78 protein, suppression of cell growth and migration, and tumorigenesis in lung cancer. Collectively, our results reveal a previously unappreciated pro-oncogenic role of OTUD3 in lung cancer and indicate that deubiquitylases could elicit tumor-suppressing or tumor-promoting activities in a cell- and tissue-dependent context.

[1] State Key Laboratory of Proteomics, National Center for Protein Sciences (Beijing), Beijing Institute of Lifeomics, 100850 Beijing, China. [2] Institute of Cancer Stem Cell, Dalian Medical University, 116044 Liaoning Province, China. [3] Institute of Systems Biomedicine, School of Basic Medical Sciences, Peking University Health Science Center, 100191 Beijing, China. [4] Beijing Key Laboratory of Chronic Heart Failure Precision Medicine, Core Laboratory of Translational Medicine, Chinese PLA General Hospital, 100853 Beijing, China. [5] Medical Isotopes Research Center and Department of Radiation Medicine, School of Basic Medical Sciences, Peking University Health Science Center, 100191 Beijing, China. [6]These authors contributed equally: Tongde Du, Hongchang Li. Correspondence and requests for materials should be addressed to C.H. (email: hanchuanchun@163.com) or to L.Z. (email: zhanglq@nic.bmi.ac.cn)

Lung cancer is the leading cause of cancer-related death worldwide with high mortality and poor prognosis[1]. Lung cancer includes small cell lung cancer (SCLC) and non-small cell lung cancer (NSCLC). NSCLC occupies 85% of the new lung cancer cases and is further divided into three categories including adenocarcinoma (ADC), squamous cell carcinoma (SCC), and large cell carcinoma through histopathological heterogeneity[2–4]. Multiple studies have demonstrated that activating mutations and ectopic expression of receptor tyrosine kinases and RAS pathway members, including KRAS, EGFR, BRAF, and HER2, would contribute to NSCLC[4]. Protein ubiquitylation is a dynamic multifaceted post-translational modification and participates in the pathogenesis of various cancers through regulating the stability or activity of certain critical proteins[5]. Recently, accumulative evidence has established the critical role of ubiquitylation in cancer pathogenesis and revealed the great therapeutic potential of targeting ubiquitylation in multiple cancers[6–9].

Like other post-translational modifications, ubiquitylation also could be reversed by peptidases termed deubiquitylases or deubiquitylating enzymes (DUBs), which could cleave and remove the ubiquitin chains from substrate proteins[10]. In the human proteome, there are about 100 DUBs consisting of six families: USPs, OTUs, UCHs, JAMMs, MJDs, and MCPIPs. Recently, we identified the OTU (ovarian tumor protease) family member OTUD3 (OTU domain-containing protein 3) is a potential tumor suppressor gene in breast cancer and OTUD3 protein directly interacts with PTEN (phosphatase and tension homologue deleted in chromosome 10) and stabilizes PTEN protein to suppress PI3K/AKT signaling transduction[11]. OTUD3 transgenic mice exhibit higher levels of the PTEN protein and are less prone to tumorigenesis of breast cancer[11]. Reduction of OTUD3 expression, concomitant with decreased PTEN abundance, correlates with human breast cancer progression[11]. Additional evidence suggested OTUD3 to be conclusively associated with ulcerative colitis in genome-wide association (GWAS) studies[12–14]. So far, current studies showed OTUD3 as a potent DUB for PTEN and then a tumor suppressor in breast cancer[11]; however, the comprehensive understandings of the role of OTUD3 in human cancers are still limited.

The glucose-regulated protein 78-kDa GRP78, also known as BiP and HSPA5, is originally identified to reside primarily in the endoplasmic reticulum (ER) of mammalian cells and control unfolded protein response (UPR) through sequestrating and maintaining the ER stress sensors including PRKR-like ER kinase (PERK), activating transcription factor 6 (ATF6) and inositol-requiring enzyme 1 (IRE1) in inactive forms[15–18]. Further studies showed that GRP78 is a multifunctional protein with activities far beyond its well-known role in the UPR and implicated in promoting tumor proliferation, metastasis and involved in drug resistance[19–25]. GRP78 could be modified with poly-ubiquitylation for subsequent degradation through the ubiquitin proteasomal system, leading to the suppression of cell migration and invasion[22,24,26,27]. Studies have demonstrated that the E3 ubiquitin ligase GP78 promotes the ubiquitylation and degradation of GRP78 and suppress tumorigenesis and metastasis[22,26].

In the present study, OTUD3 knockout mice were generated and crossed with spontaneous breast cancer mice (MMTV-PyMT mice) and inducible NSCLC mice (Kras G12D mice), and we find that OTUD3 deletion results in increased susceptibility to breast cancer, but decreased susceptibility to NSCLC. Further tissue microarray analysis shows that the expression levels of OTUD3 are decreased, concomitant with reduction of PTEN abundance, in human breast cancer, hepatocellular cancer, colon cancer, and cervical cancer. Strikingly, OTUD3 is upregulated in human lung cancer and elevated expression of OTUD3 is associated with poor prognosis in lung cancer patients. Mechanistically, OTUD3 promotes tumorigenesis of the lung adenocarcinoma through deubiquitylating and stabilizing GRP78. These results reveal GRP78 as a substrate of OTUD3 deubiquitylase and broaden the understanding of physiological tumor-associated function of OTUD3 in multiple types of human cancer.

## Results

**Deletion of OTUD3 promotes breast cancer but inhibits lung cancer development.** Our previous data demonstrated that OTUD3 acts as a tumor suppressor in breast cancer by maintaining PTEN stability and OTUD3 transgenic (TG) mice are less prone to tumorigenesis of breast cancer[11]. To further investigate the physiological and pathological functions of OTUD3 in vivo, Loxp-Cre strategy-mediated global deletion of the OTUD3 was introduced into mice (Supplementary Fig. 1a, b). Homozygous $OTUD3^{-/-}$ mice were born at the expected Mendelian frequency and western blot analysis of mouse embryonic fibroblasts (MEFs) and tissues from OTUD3 knockout (KO) mice and the wild-type (WT) littermates confirmed the successful deletion of OTUD3 protein (Supplementary Fig. 1c–e). We went on to detect the PTEN protein levels in OTUD3 KO mice and found PTEN levels in WAT (white adipose tissue) and muscle of OTUD3 KO mice were markedly decreased, whereas the protein levels of PTEN in other tissues examined were comparable between the WT and KO groups (Supplementary Fig. 1f). Consistently, the level of phosphorylated AKT was higher in OTUD3 KO WAT and muscle tissues than that in WT tissues (Supplementary Fig. 1f). Moreover, spontaneous tumor formation can scarcely be detected in OTUD3 KO mice within one year of age under normal feed conditions (data not shown). To unravel the physiological roles of OTUD3 in multiple typical cancers, we crossed the OTUD3 KO and WT mice with MMTV-polyomavirus middle T antigen (PyMT) transgenic mice and $Kras^{LSL-G12D/WT}$ mice, separately. As expected, all of PyMT/OTUD3 WT mice spontaneously developed breast tumors at 80–82 days after birth. Strikingly, the earliest tumor lumps in PyMT/OTUD3 KO mice appears at 70–72 days and more than 50% of the mice developed tumors at 80 days (Fig. 1a, b). The volume and weight of tumor tissues from PyMT/OTUD3 KO mice were significantly bigger than those of PyMT/OTUD3 WT mice (Fig. 1c–f). Compared to breast cancer tissues of PyMT/OTUD3 WT mice, decreased PTEN protein level and increased p-AKT (Ser-473) level were observed in breast cancer tissues of PyMT/OTUD3 KO mice (Fig. 1g and Supplementary Fig. 1g). The PyMT/OTUD3 KO mice showed enhanced rate of cell proliferation, angiogenesis, and dampened rate of cell apoptosis in tumor formation compared with that of the the PyMT/OTUD3 WT mice, as indicated by the Ki67, CD31 and cleaved-Caspase-3 staining (Fig. 1g). These results suggest that OTUD3 is a tumor suppressor in breast cancer in vivo.

Next, to unveil the role of OTUD3 in $Kras^{G12D}$-driven lung tumor formation, the $Kras^{LSL-G12D/WT}/OTUD3$ WT and $Kras^{LSL-G12D/WT}/OTUD3$ KO mice were obtained (Supplementary Fig. 1h). Adeno-associated virus expressing Cre-recombinase-GFP (AAV-Cre-GFP) were delivered into $Kras^{LSL-G12D/WT}/OTUD3$ WT and $Kras^{LSL-G12D/WT}/OTUD3$ KO mice to induce expression of Kras$^{G12D}$ protein. The efficiency of AAV-mediated gene delivery was evaluated by bioluminescence imaging and equal introduction of GFP-Cre signals were monitored to exhibit in the lung (Supplementary Fig. 1i). We then analyzed the survival ratio between $Kras^{G12D/WT}/OTUD3$ WT and $Kras^{G12D/WT}/OTUD3$ KO mice. Surprisingly, we found that genetic ablation of OTUD3 resulted in a markedly prolonged survival (Fig. 1h). Moreover, we imaged a separate cohort of $Kras^{G12D/WT}/OTUD3$ WT and $Kras^{G12D/WT}/OTUD3$ KO mice at 8 and 12 weeks post AAV-Cre infection using Micro-CT (Micro-Computed Tomography)

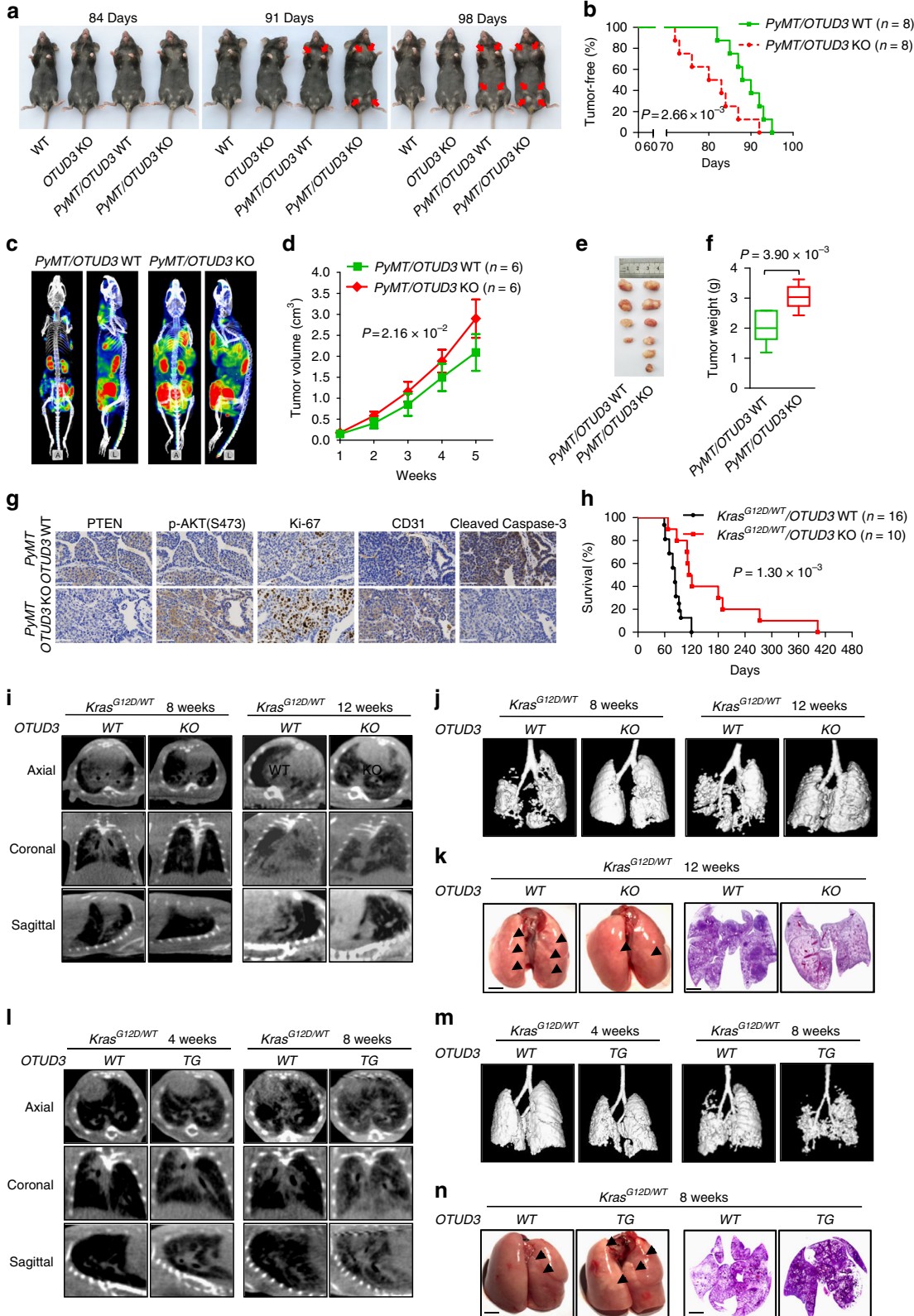

imaging. $Kras^{G12D/WT}/OTUD3$ KO mice showed less detectable nodules than $Kras^{G12D/WT}/OTUD3$ WT mice in the lung by coronal and transverse section imaging μCT scan (Fig. 1i, j). Subsequently, we performed tumor analysis in mice at 12 weeks post AAV-Cre-induction. $Kras^{G12D/WT}/OTUD3$ KO mice exhibited a dramatically decreased tumor burden, tumor area, which correlated with lower tumor grades compared with that of the

$Kras^{G12D/WT}/OTUD3$ WT mice (Fig. 1k and Supplementary Fig. 1j–l). Importantly, tumor sections from $Kras^{G12D/WT}/OTUD3$ KO mice exhibited decreased cell proliferation, shrinked angiogenesis, and accumulated cell apoptosis signals than that of tumor tissues from $Kras^{G12D/WT}/OTUD3$ WT mice, as indicated with the Ki67, CD31, and cleaved Caspase-3 staining (Supplementary Fig. 1m).

**Fig. 1** Comparison of tumorigenesis in *OTUD3* WT and KO mice. **a** Representative bright-field imaging of mammary adenocarcinoma from WT, *OTUD3* KO, *PyMT/WT*, and *PyMT/OTUD3* KO littermates and arrows indicate surface tumors. **b** Mammary adenocarcinoma incidence in *PyMT/WT* ($n = 8$) and *PyMT/OTUD3* KO ($n = 8$) littermates depicted as the percentage of tumor-free mice. Mice were considered to be tumor free until a palpable mass (>4.0 mm) persisted for longer than 4 days. Log-rank test. **c** Representative whole-body posterior and left lateral images of *PyMT/WT* and *PyMT/OTUD3* KO littermates with breast tumors obtained from single-photon emission computed tomography (SPECT)/CT imaging at 0.5 h after $^{99m}$Tc-HHK injection. **d** Total tumor volume of *PyMT/WT* ($n = 6$) and *PyMT/OTUD3* mice ($n = 6$) were analyzed once a week after tumor formation. Two-way ANOVA test. **e**, **f** The tumor at the fifth week of indicated mice were obtained and calculated (*PyMT/WT*, $n = 6$; *PyMT/OTUD3*, $n = 6$). Mann–Whitney test. **g** Immunohistochemistry analysis protein levels of PTEN, p-AKT(S473), Ki-67, CD31, and cleaved Caspase-3 (scale bar, 100 μm) in breast tumors of *PyMT/WT* and *PyMT/OTUD3* KO littermates, respectively. **h** Kaplan–Meier plot showing overall survival of mice with the indicated genotypes infected with AdCre (*Kras$^{G12D/WT}$/WT*, $n = 16$; *Kras$^{G12D/WT}$/OTUD3* KO, $n = 10$), log-rank test. **i**, **l** Micro-CT images in indicated planes from *OTUD3* WT or KO mice (**i**) and *OTUD3* WT or TG mice (**l**) indicated weeks post-infection ($1 \times 10^{10}$pfu AAV-GFP-Cre). **j**, **m** Three-dimensional rendering of micro-CT data with lungs in gray, lost part represented tumor. **k**, **n** Representative images of lung lesions (**k**, **n**, left, scale bar, 0.5 cm) and H&E staining (**k**, **n**, right, scale bar, 0.5 cm) from the indicated experimental group and arrows indicate surface tumors. For the box and whisker graphs in **f**, the box extends from the twenty-fifth to the seventy-fifth percentile, the line inside the box is the median and the whiskers are the fifth and ninety-fifth percentiles. Statistics source data can be found in Supplementary Data 1

To further confirm the carcinogenic function of OTUD3 in lung cancer in vivo, OTUD3 transgenic (OTUD3 TG) mice with overexpression of OTUD3 protein, were crossed with *Kras$^{LSL-G12D/WT}$/OTUD3* WT mice to obtain *Kras$^{LSL-G12D/WT}$/OTUD3* TG mice (Supplementary Fig. 2a, b). Following analysis demonstrated that *Kras$^{G12D/WT}$/OTUD3* TG mice displayed more detectable nodules, larger tumor area, higher tumor grades than *Kras$^{G12D/WT}$/OTUD3* WT mice in the lung and exhibited poorer overall survival than those bearing tumors with normal expression level of OTUD3 (Fig. 1l–n and Supplementary Fig. 2c–f). Moreover, the tumor tissues from *Kras$^{G12D/WT}$/OTUD3* TG mice have faster cell proliferation, angiogenesis, and slower cell apoptosis rate as indicating with the Ki67, CD31, and cleaved Caspase-3 staining (Supplementary Fig. 2g). Collectively, we concluded that OTUD3 plays distinctive roles in different types of tumors. Notably, OTUD3 functions as a tumor promoter in the lung adenocarcinoma in vivo instead of a tumor suppressor.

**OTUD3 is a positive indicator of NSCLC.** Previous study has indicated that the expression level of OTUD3 was downregulated in human breast cancer and was notably associated with PTEN expression[11]. To further validate the different roles of OTUD3, we firstly evaluated the correlation between OTUD3 and PTEN in hepatocellular cancer ($n = 73$), colon cancer ($n = 71$), cervical cancer ($n = 30$), and lung adenocarcinoma ($n = 85$). Consistent with our previous finding in breast cancer, the expression levels of OTUD3 and PTEN were decreased in hepatocellular cancer, colon cancer, and cervical cancer, compared with the matched adjacent normal tissues, and the expression of OTUD3 was closely related to PTEN expression in hepatocellular cancer, colon cancer, and cervical cancer (Fig. 2a–i). In contrast, in lung adenocarcinoma, the expression level of OTUD3 was increased as compared with the matched adjacent normal tissues and it was irrelevant with the expression of PTEN, whose expression was reduced in lung adenocarcinoma (Fig. 2j–l). Similar result was obtained in lung squamous carcinoma (Supplementary Fig. 2h, i). Furthermore, we also found that patients bearing lung tumors with relatively high levels of OTUD3 showed poorer overall survival than those bearing tumors with low OTUD3 (Supplementary Fig. 2j). These findings suggest the notion that OTUD3 might be a tumor promoter in the lung adenocarcinoma.

**OTUD3 suppresses tumorigenicity and metastasis in colon, liver, and cervical cancer.** In consideration of the opposite expression of OTUD3 in different types of cancer, we next used human breast cancer cell MCF7, MDA-MB-231, hepatocellular cancer cell HepG2, colon cancer cell HCT116, and cervical cancer cell HeLa to further investigate the functions of OTUD3 in tumorigenesis and tumor metastasis. Depletion of OTUD3 dramatically decreased PTEN levels, and two independent OTUD3 shRNAs showed significant effects (Supplementary Fig. 3a, b). Moreover, ectopic expression of OTUD3 resulted in PTEN protein elevation in a dose-dependent manner in these cells (Supplementary Fig. 3c). We next investigated the role of OTUD3 on cell proliferation and migration. Depletion of OTUD3 in those three cell lines promoted cell proliferation (Fig. 3a) and induced anchorage-independent growth (Fig. 3b). To further assess whether OTUD3 is a bona fide tumor suppressor gene, we sought to determine whether loss of OTUD3 expression would induce tumorigenicity. Depletion of OTUD3 in the examined cells all resulted in promotion of tumor growth in xenografted nude mice (Fig. 3c, d). We also investigated whether OTUD3 knockdown would affect cell migration and tumor metastasis. Stable depletion of OTUD3 enhanced the migration ability of HCT116, HepG2, and HeLa cells to about three folds (Fig. 3e). The OTUD3-depleted or control cells were subcutaneously implanted into the nude mice. Six weeks later, we found that transferred liver tumors were formed in 60% (3/5) of mice injected with OTUD3-depleted cells, but not those with control cells (Fig. 3f, g). Furthermore, we utilized the lateral tail vein injection to innoculate the cells into the nude mice, and six weeks later we observed lung tumor formation in 40–80% of mice injected with OTUD3-depleted cells, but not in the control groups (Fig. 3h, i). Collectively, these results suggest that OTUD3 is a bona fide suppressor of tumor growth and metastasis events in colon cancer, liver cancer, and cervical cancer. However, when we further investigated whether OTUD3 stabilizes PTEN in lung cancer cell H1299 and A549, we found that OTUD3 failed to maintain PTEN level (Supplementary Fig. 3d, e), The seemingly contradictory phenomenon implied us that there is an unrevealed mechanism and function, which is independent of PTEN, of OTUD3 in lung cancer regulation.

**OTUD3 facilitates tumorigenicity of human lung cancer.** To further investigate the role of OTUD3 in human lung cancer, we knocked down endogenous OTUD3 in H1299 and A549 cells. Compared with the control cells, OTUD3 depletion suppressed cell growth and migration (Fig. 4a, d and supplementary Fig. 4a, c). Since OTUD3 is a deubiquitylase, we asked whether the promotion of OTUD3 in NSCLC of tumorigenicity was dependent on its enzyme activity. The catalytically inactive mutant OTUD3 C76A was generated and then the OTUD3-depleted H1299 and A549 cells ectopically expressing WT or C76A OTUD3 were established. We found overexpression of WT but

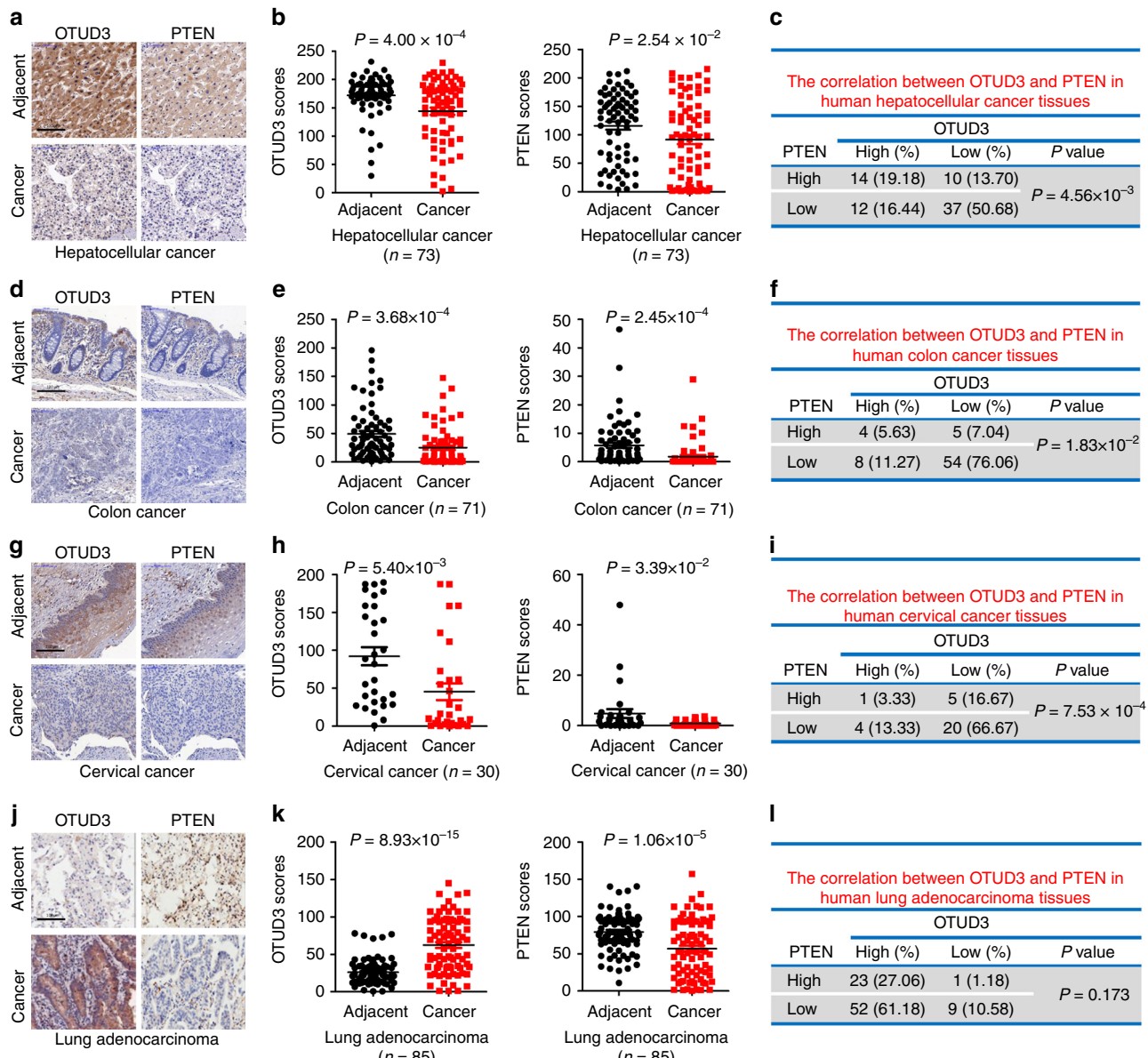

**Fig. 2** Analysis of the association between OTUD3 and PTEN in cancer. **a** Representative images from immunohistochemical staining of OTUD3 and PTEN in hepatocellular carcinoma ($n = 73$) and matched adjacent tissue (scale bar, 100 μm). **b** OTUD3 and PTEN protein levels in cancer tissues are compared with matched adjacent normal tissues in hepatocellular carcinoma by using a Wilcoxon matched pairs test. **c** The correlation of OTUD3 with PTEN protein levels was analyzed in Human Hepatocellular Cancer Tissues. **d** Representative images from immunohistochemical staining of OTUD3 and PTEN in Colon cancer ($n = 71$) and matched adjacent tissue (scale bar, 100 μm). **e** OTUD3 and PTEN protein levels in cancer tissues are compared with matched adjacent normal tissues in Colon cancer by using a Wilcoxon matched pairs test. **f** The correlation of OTUD3 with PTEN protein levels was analyzed in Human Colon Cancer Tissues. **g** Representative images from immunohistochemical staining of OTUD3 and PTEN in Cervical carcinoma ($n = 30$) and matched adjacent tissue (scale bar, 100 μm). **h** OTUD3 and PTEN protein levels in cancer tissues are compared with matched adjacent normal tissues in Cervical carcinoma by using a Wilcoxon matched pairs test. **i** The correlation of OTUD3 with PTEN protein levels was analyzed in Human Cervical Cancer Tissues. **j** Representative images from immunohistochemical staining of OTUD3 and PTEN in Lung adenocarcinoma ($n = 85$) and matched adjacent tissue (scale bar, 100 μm). **k** OTUD3 and PTEN protein levels in cancer tissues are compared with matched adjacent normal tissues in Lung adenocarcinoma by using a Wilcoxon matched pairs test. **l** The correlation of OTUD3 with PTEN protein levels was analyzed in Human Lung Lung Adenocarcinoma Tissues. The samples were classified into two groups (low OTUD3, high OTUD3) based on the OTUD3 level relative to the scores of the whole cohort. Data were analysed using Mann–Whitney test (**c**, **f**, **i**, **l**). Statistics source data can be found in Supplementary Data 1

not C76A OTUD3 facilitated cell growth and migration (Fig. 4b, e and Supplementary Fig. 4b, d). Similar results were obtained in human normal bronchial epithelium HBE cells (Fig. 4c).

To further validate the tumor promotion roles of OTUD3 in vivo, we evaluated the effect of OTUD3 on tumor growth in nude mice and found OTUD3 knockdown shrinked tumor growth (Fig. 4f, h and Supplementary Fig. 4e, f) whereas OTUD3

overexpression accelerated tumor growth in vivo (Fig. 4g, i and Supplementary Fig. 4g, h). Collectively, our data indicated that OTUD3 plays an oncogenic role in lung cancer cells and its regulation of tumor growth is indispensable on its DUB activity.

**OTUD3 interacts with GRP78.** To uncover the molecular mechanisms whereby OTUD3 promoted tumorigenicity of lung

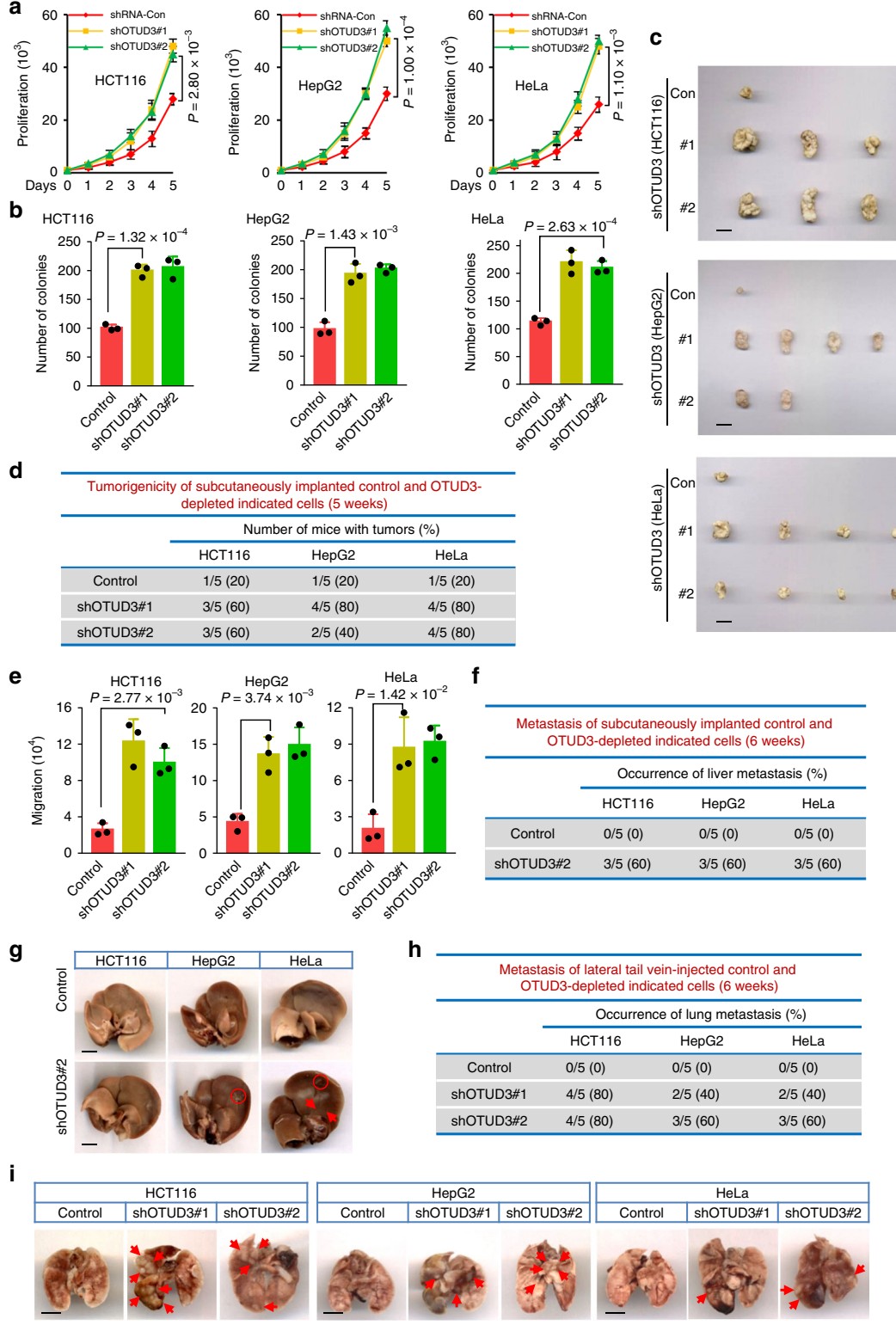

cancer, mass spectrometry analysis was exerted to discover the OTUD3 interacting partners (Supplementary Data 2). As shown in Supplementary Fig. 5a, the potential interacting proteins with high fidelity were listed. Trim25, YB1, and GRP78, which played important roles in tumorigenesis, guided us for further examination[28–31]. To confirm the physical associations of OTUD3 with these putative binding partners, exogenous co-immunoprecipitation (Co-IP) assays were performed in cultured cells. Among the three proteins examined, GRP78 was the

only reproducible interactor (Fig. 5a, b and Supplementary Fig. 5b–d). Moreover, purified GST-OTUD3 protein, but not the GST alone, was able to interact with His-GRP78 under cell-free conditions (Fig. 5c), suggesting a direct interaction between GRP78 and OTUD3. Whereafter, the interaction between endogenous OTUD3 and GRP78 was detected in A549 and H1299 cells (Fig. 5d). GRP78 is a HSP70 molecular chaperone that binds newly synthesized proteins and maintains them in a state competent for subsequent folding and oligomerization[32]. Since

**Fig. 3** OTUD3 suppresses tumorigenicity and tumor metastasis. **a** Cell proliferation was measured in the indicated cells stably expressing the shRNA-Control, shOTUD3#1 and shOTUD3#2. The data are shown as mean ± s.d. $n = 3$ independent experiments, two-way ANOVA test. **b** The indicated cells were tested for anchorage-independent growth in a soft-agar colony assay, viable colonies after 3 weeks were counted. Data are depicted as bar graphs with mean ± s.d. $n = 3$ independent experiments. Student's $t$-test. **c** OTUD3 shRNA- HCT116, HepG2, and HeLa cells were subcutaneously injected into nude mice for tumor formation ($2 \times 10^6$ cells per mouse, 5 weeks) assays (scale bar, 0.5 cm). Representative bright-field imaging of the tumors from the sacrificed mice implanted indicated HCT116, HepG2, and HeLa cells stably expressing OTUD3 shRNA or control shRNA. **d** On 5 weeks, mice receiving transplants of indicated cells were sacrificed and mice forming tumors were analysed. **e** Transwell assay of the OTUD3 shRNA-transduced HCT116, HepG2 and HeLa cells. Data are depicted as bar graphs with mean ± s.d. $n = 3$ independent experiments; Student's $t$-test. **f** The indicated cells were subcutaneously injected into 6-week-old nude mice ($2 \times 10^6$ cells per mouse), and the liver metastasis was analyzed after 6 weeks. **g** Representative bright-field imaging of the liver metastasis (scale bar, 0.5 cm). Rings and arrows indicate surface metastatic nodules. **h** $2 \times 10^6$ cells from indicated cell lines were injected per mouse into 6-week-old nude mice through lateral tail vein. Each cell line was injected in five different animals, and lung metastasis was analyzed after 6 weeks. **i** Representative bright-field imaging of the lung metastasis (scale bar, 0.5 cm). Arrows indicate surface. Statistics source data can be found in Supplementary Data 1

OTUD3 is a classical member of the OTU family of the deubiquitylase, we wondered whether OTUD3 specifically interacted with GRP78. The interactions between GRP78 and additional OTUs were examined. As shown in Fig. 5e, f and Supplementary Fig. 5d, among the examined nine OTUs, only OTUD3 was able to bind to GRP78. Additionally, mapping the region of GRP78 required for OTUD3 binding revealed that the C-terminal region (aa 500-654) of GRP78 was critical for interaction with OTUD3 (Fig. 5g and Supplementary Fig. 5e). Similarly, the N-terminal OTU region of OTUD3 mediated the physical interaction with GRP78 (Fig. 5h and Supplementary Fig. 5f). Collectively, these results demonstrated that GRP78 is a bona fide interacting protein for OTUD3.

**OTUD3 deubiquitylates GRP78 and maintains GRP78 stability**. To determine the possible regulatory effect of OTUD3 on GRP78, WT or C76A OTUD3 were overexpressed in two lung cancer cell lines. We found that WT OTUD3, but not C76A mutant, increased GRP78 protein level (Fig. 6a and Supplementary Fig. 6a) whereas OTUD3 knockdown remarkably decreased GRP78 protein level (Fig. 6b and Supplementary Fig. 6b). This effect was restored by treatment with the proteasome inhibitor MG132 (Fig. 6c and Supplementary Fig. 6c) or expression of a shRNA-resistant OTUD3 (Fig. 6d). To further validate that OTUD3 affects GRP78 protein stability, we treated indicated cells with the protein synthesis inhibitor cycloheximide (CHX). Notably, overexpression of OTUD3 led to a prolonged half-life of endogenous GRP78 protein (Fig. 6e), whereas depletion of OTUD3 resulted in a shortened half-life of GRP78 protein (Fig. 6f).

OTUD3 is a deubiquitylase that could remove ubiquitin chain from its substrate[11,33]. As OTUD3 binds to and positively regulate GRP78 protein stability, we hypothesized that OTUD3 might increase GRP78 expression via deubiquitylation of GRP78. We found that GRP78 can be ubiquitylated in lung cancer cells and ectopic expression of WT, but not C76A OTUD3, reduced GRP78 ubiquitylation in cells (Fig. 6g, h). Conversely, knockdown of OTUD3 increased GRP78 ubiquitylation (Fig. 6i) and this effect was reversed by introduction of shRNA-resistant OTUD3 (Fig. 6j).

Our data have indicated that OTUD3 fails to regulate PTEN stability in lung cancer whereas casting its role in mediating GRP78 deubiquitylation. To further verify our findings, we tested whether OTUD3 can regulate GRP78 stability in other types of cancer cells. Surprisingly, we found that overexpression of OTUD3 did not alter the stability or ubiquitylation level of GRP78 in MCF7 cells and HeLa cells (Supplementary Fig. 6d–i). The interaction between OTUD3 and GRP78 observed in lung cancer H1299 cells was hardly detected in MCF7 cells (Supplementary Fig. 6j,k). Since GRP78 is a protein localized mainly in the ER and PTEN is mainly localized in the cytoplasm but barely in the ER, OTUD3 may show the different subcellular localization in different cell types. To investigate it, we next analyzed the colocalization of OTUD3 and GRP78 or PTEN in lung and breast cancer cell lines using immunofluorescence and used Calnexin as a ER marker. We found that OTUD3 colocalized with GRP78 in the cytoplasm (especially in ER) but not colocalized with PTEN in lung cancer cells (Supplementary Fig. 6l, m). Whereas, in breast cancer cells OTUD3 was hardly detected to be colocalized with GRP78 in the ER (Supplementary Fig. 6n). Taken together, these data suggest that GRP78 is a specific substrate of OTUD3 in lung cancer cells.

**OTUD3 accelerates lung cancer cell growth and metastasis via regulating GRP78**. To evaluate whether OTUD3 promotes tumor progression through regulating GRP78, we depleted endogenous GRP78 in OTUD3-overexpressing H1299 and A549 cells and observed that the inhibition of endogenous GRP78 expression attenuated OTUD3-induced cell growth and migration (Fig. 7a–f). Consistently, xenograft assays showed that depletion of GRP78 rescued the effects of OTUD3 overexpression and inhibited the tumor development (Fig. 7g–i). Collectively, these results demonstrate that OTUD3 facilitated tumor growth and metastasis by regulating GRP78.

The data in Fig. 2 showed that OTUD3 expression was significantly increased in human lung cancer tissues and had no significant correlation with PTEN. We then investigated the correlation between OTUD3 and GRP78 in human lung cancer tissues. Immunohistochemistry analysis of a tumor microarray ($n = 112$) showed that about 78.6% of samples with high OTUD3 samples exhibited high GRP78 expression parallelly (Fig. 7j, k). Additionally, we also assessed the correlation between OTUD3 and GRP78 in nude mice introduced by H1299 and A549 cells with OTUD3 overexpression (Supplementary Fig. 7a) or knockdown (Supplementary Fig. 7b). Expectedly, the in situ expression of GRP78 was elevated in OTUD3 overexpressing group while declined in OTUD3-depleted groups (Supplementary Fig. 7a, b). In consistent with these phenomenon, elevated OTUD3 expression increased GRP78 protein level in the lung tissue of OTUD3 TG mice (Supplementary Fig. 7c, d), whereas OTUD3 knockout decreased GRP78 expression in the mouse lung tissue (Supplementary Fig. 7e, f). To sum up, these data highlighted the function and mechanism of OTUD3 to facilitate lung tumor growth.

**Discussion**

Deubiquitylase OTUD3 has been identified to suppress PI3K/AKT signalling via upregulating PTEN protein level[11]. Although evidence based on *OTUD3* transgenic mice model showed the

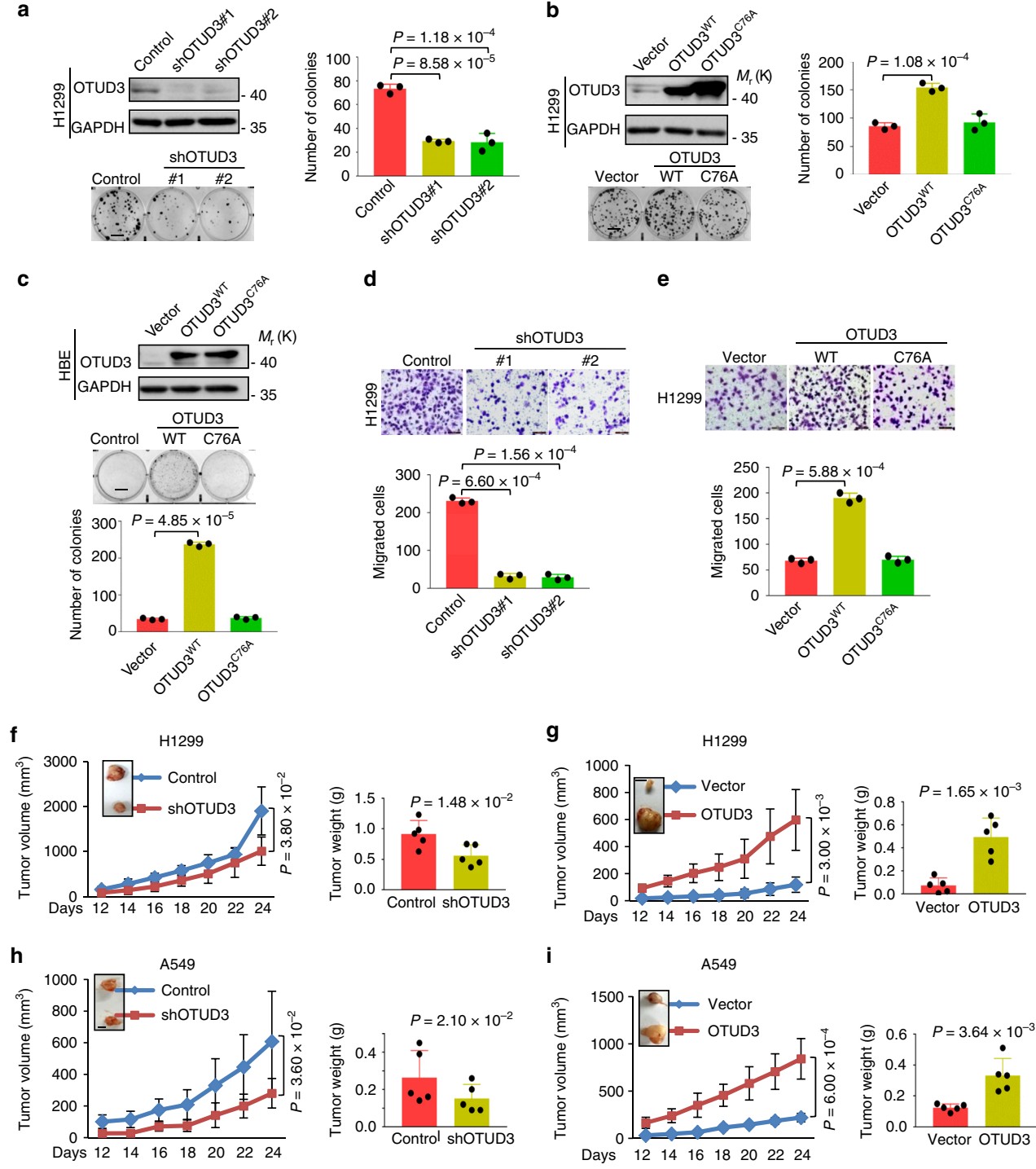

tumor suppressive role of OTUD3 in breast cancer, whether deletion of OTUD3 in vivo could promote the occurrence and development of breast cancer remains mysterious. In this study, *OTUD3* knockout mice were firstly generated and utilized to evaluate the physiological role of OTUD3 in tumorigenesis. As expected, deficiency of OTUD3 accelerated the development of mammary tumor in *MMTV-PyMT* transgenic mice, which was consistent with our previous conclusion in *OTUD3* transgenic mice. Surprisingly, the completely opposite finding was obtained in *Kras^{LSL-G12D/t}/OTUD3* KO mice that deletion of OTUD3 decreased rather than increased lung tumor development. Thus, these data indicated the pathological function of OTUD3 in tumorigenesis was differential in different tissues.

Several lines of evidence support the concept that OTUD3 exhibits tumor-promoting effect in lung cancer. Firstly, the expression level of OTUD3 was significantly upregulated in lung cancer tissues compared with adjacent tissues, and the positive correlation between OTUD3 and PTEN detected in breast cancer, hepatocellular cancer, colon cancer, cervical cancer, disappeared in human lung cancer. Secondly, although OTUD3 can inhibit tumor growth and metastasis by stabilizing PTEN in breast cancer, hepatocellular cancer, colon cancer, and cervical cancer, loss of OTUD3 suppressed the tumorigenesis of lung cancer in vitro and in vivo. What is noteworthy is that OTUD3 promotes tumorigenesis both in tumor cells and in non-tumor cells within the microenvironment (for example, tumor stroma cell). In order

**Fig. 4** OTUD3 facilitates tumorigenicity in human lung cancer cells. **a** OTUD3 was knocked down in lung cancer cell lines H1299 using shRNA. The protein levels of OTUD3 were analyzed by western blotting. Effects of OTUD3 on the cell growth were examined by colony formation assay (scale bar, 1 cm). Results are representative of $n = 3$ independent experiments. **b** OTUD3 WT and C76A were overexpressed in H1299 cells. The protein levels of OTUD3 WT and OTUD3 C76A were analyzed by western blotting. Effect of OTUD3 WT or C76A on the H1299 cells growth was examined by colony formation (scale bar, 1 cm). Results are representative of $n = 3$ independent experiments. **c** OTUD3 WT or OTUD3 C76A were overexpressed in human normal bronchial epithelium cells HBE. The protein levels of OTUD3 WT and OTUD3 C76A were analyzed by western blotting. Effects of OTUD3 WT or C76A on the cell growth were examined by colony formation assay (scale bar, 1 cm). Results are representative of $n = 3$ independent experiments. **d** OTUD3 was knocked down in lung cancer cell lines H1299 using shRNA. Effects of OTUD3 on the migration were examined by transwell assay (scale bar, 100 μm). Migrated cells were plotted as the average number of cells per field of view. Results are representative of $n = 3$ independent experiments. **e** Effect of OTUD3 WT or C76A on the migration were examined by transwell assay (scale bar, 100 μm). Migrated cells were plotted as the average number of cells per field of view. Results are representative of $n = 3$ independent experiments. **f** H1299 cells with or without OTUD3 knockdown were subcutaneously injected ($2 \times 10^6$ cells per mouse) into 6-week-old nude mice ($n = 5$). Tumor volume and weights were analysed (scale bar, 0.5 cm). **g** H1299 cells with or without overexpressing OTUD3 were subcutaneously injected ($1 \times 10^6$ cells per mouse) into 6-week-old nude mice ($n = 5$). Tumor volume and weights were analysed (scale bar, 0.5 cm). **h** A549 cells with or without OTUD3 knockdown were subcutaneously injected ($2 \times 10^6$ cells per mouse) into 6-week-old nude mice ($n = 5$). Tumor volume and weights were analysed (scale bar, 0.5 cm). **i**, A549 cells with or without overexpressing OTUD3 were subcutaneously injected ($1 \times 10^6$ cells per mouse) into 6-week-old nude mice ($n = 5$). Tumor volume and weights were analysed (scale bar, 0.5 cm). For **a**–**e**, the data are shown as mean ± s.d. Student's $t$-test. For panels **f**–**i**, tumor volume was analyzed by two-way ANOVA test, tumor weight was analyzed by Student's $t$-test. Statistics source data can be found in Supplementary Data 1. Uncropped images of blots are shown in Supplementary Fig. 9

to assess whether OTUD3 could influence tumorigenesis in the tumor stroma, a mouse lung cancer cell line LLC was used to orthotopically transplant into WT or *OTUD3* KO mice, as well as WT or *OTUD3* TG mice. The results showed that deficiency of OTUD3 suppressed the tumorigenic ability of LLC cells in the lung (Supplementary Fig. 8a–d), and overexpression of OTUD3 enhanced the tumorigenic ability of LLC cells in the lung (Supplementary Fig. 8e–h). These data suggest that OTUD3 not only plays a oncogenic role in lung cancer cells, but also promotes the development of lung cancer by affecting the growth environment of lung cancer cells. Finally, the interaction between OTUD3 and PTEN was not observed in lung cancer cells, and OTUD3 turned to interact with another substrate GRP78, a regulator which could promote tumorigenesis, metastasis and drug resistance in cancers[34–37]. In general, we revealed that OTUD3 functions as a tumor promoter in lung cancer by maintaining the protein level of GRP78, whereas exhibits tumor suppressive role through stabilizing PTEN in breast cancer, hepatocellular cancer, colon cancer, and cervical cancer.

GRP78 is frequently overexpressed in numeral kinds of human cancers[23,38–41], although ubiquitin ligase GP78 has been discovered to downregulate the protein stability of GRP78 and suppress tumor metastasis[42], the deubiquitylase which stabilizes GRP78 protein level has not been defined. We identified OTUD3 as the first deubiquitylase of GRP78. OTUD3 maintained GRP78 stability, leading to lung cancer cell growth and tumorigenesis in vitro and in vivo, even though the precise mechanisms that OTUD3 plays opposite role in different cancers and elevate expression in lung cancer still need further investigation. Surprisingly, we found that under physiological conditions, the levels of PTEN protein in most tissues of OTUD3 knockout mice showed no obvious difference from those of wild-type mice. We hypothesize that the PTEN-protection activity of OTUD3 might only be initiated in those already established cancerous tissues or cells. In addition, considering both OTUD3 and USP13 were reported to remove polyubiquitin chain of PTEN and synergistically stabilize PTEN[11], generation of the *OTUD3/USP13* double knockout mice would be a better way to investigate the physiological role of OTUD3 and USP13 in regulating PTEN.

Although the exact mechanism is unclear so far, a recent study pointed out that a gene family members can make breast cells proliferate like cancer cells, but not in other tissue-type of cells, even inhibit cell proliferation[43]. That means some genes play a role in the function of tumor suppressor or carcinogenesis may rely on tissue specificity, and an effective cancer treatment

strategy in an organization does not guarantee to be effective in other tissues and may even be counterproductive. In our study, we describes a PTEN protector OTUD3 discards the regulation of PTEN and turns to maintain the stability of an oncoprotein GRP78 in lung cancer, and deletion of OTUD3 in mice decreased development of lung carcinomas, implicating the tissue complexity of deubiquitylase in the development of tumor. This finding also implies it should be considered that developing tumor therapy by targeting deubiquitylases in specific organ or tissue may induce tumor in another one.

## Methods

**Mouse model.** The *OTUD3* knockout mouse model was generated by Model Animal Research Center of Nanjing University. Strategy of *OTUD3* knockout mouse model was illustrated in Supplementary Fig. 1a. After Cre/loxP excision, frame-shift caused by lack of exon 2 produces stop codes in exon 3 and stop the translation prematurely and destroy the later residues, the gene product is eliminated; *OTUD3* knockout mice were identified by PCR (polymerase chain reaction) of tail-tip genomic DNA. Amplification of the *OTUD3* WT and KO mice results in 508 bp and 470 bp PCR products, respectively. The *OTUD3* transgenic mice were generated by Cyagen Biosciences and the strategy was described in our previous study[11]. Transgenic mice were identified by PCR of tail-tip genomic DNA and amplification of the transgene is a 528 bp PCR product. To obtain the *PyMT/OTUD3* KO mice, *MMTV-PyMT* mice (*PyMT/WT* mice, FVB/N background) were crossed with *OTUD3* KO mice. In order to exclude the genetic background on the experimental results, an indicated *PyMT/OTUD3*<sup>WT/-</sup> male mouse, which was the first generation offspring of *PyMT/WT* cross with *OTUD3* KO mice, was used to cross with WT and *OTUD3* KO mice, respectively, and then the *PyMT/OTUD3* WT and *PyMT/OTUD3* KO mice were obtained from the offspring through PCR screen. The amplification of the *MMTV-PyMT* mice is a 556 bp PCR product. For spontaneous mammary tumor experiment, female *WT/OTUD3* WT ($n = 8$), *WT/OTUD3* KO ($n = 8$), *PyMT/OTUD3* WT ($n = 8$) and *PyMT/OTUD3* KO ($n = 8$) mice were used to observe breast carcinogenesis from 8 weeks old. Tumor size was measured every week after breast carcinogenesis by Vernier caliper and converted to TV according to the following formula: TV (mm$^3$) = (a × b$^2$)/2, where a and b represent the largest and smallest diameters, respectively. For single-photon emission computed tomography (SPECT)/CT imaging, a paired of *PyMT/OTUD3* WT and *PyMT/OTUD3* KO litters (14 weeks) were injected with <sup>99m</sup>Tc-HHK through lateral tail vein, then mice were narcotized after half an hour and used for NanoSPECT imaging. On the day of 5 weeks after breast carcinogenesis, all of the female *PyMT/OTUD3* WT and *PyMT/OTUD3* KO mice were subjected to euthanasia and the spontaneous mammary tumors were taken out for experimental comparison. *Kras*<sup>LSL-G12D/WT</sup> mice (C57BL/6 J background) were obtained from Shanghai Model Organisms Center, Inc and used to generate *Kras*<sup>LSL-G12D/WT</sup>/ *OTUD3* KO and *Kras*<sup>LSL-G12D/WT</sup>/*OTUD3* TG mice through crossing with *OTUD3* KO mice and *OTUD3* TG mice respectively. *Kras*<sup>LSL-G12D/WT</sup> mice were identified by PCR of tail-tip genomic DNA. The PCR amplifications of the *Kras*<sup>LSL-G12D/WT</sup> mice are 250 bp and 100 bp, whereas the PCR product of wild-type mice is only 250 bp. All of the primers for genotyping are reported in Supplementary Table 1. All experimental procedures in mice were approved by the Laboratory Animal Center of Chinese Academy of Military Medical Sciences and complied with all relevant ethical regulations.

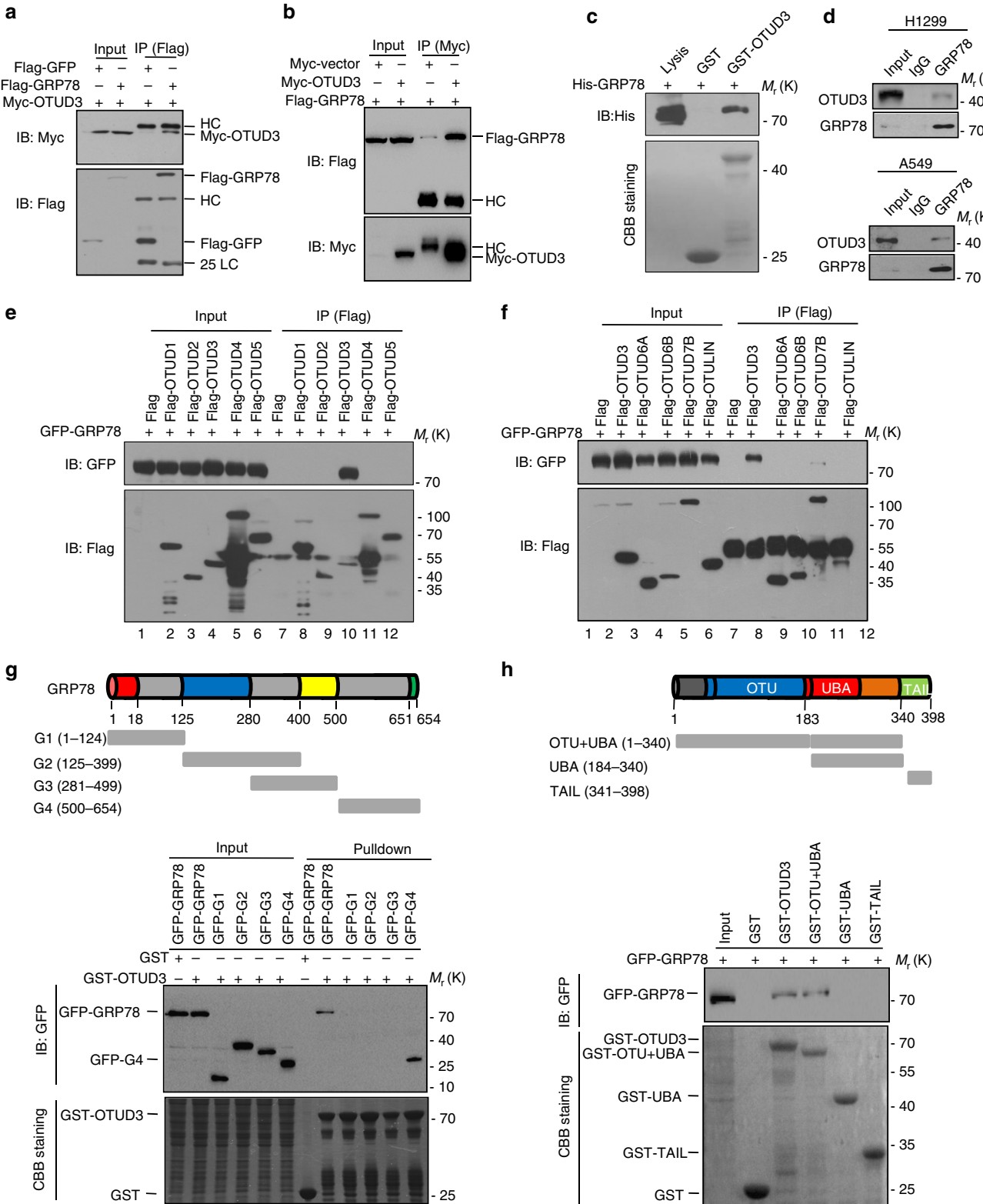

**Fig. 5** OTUD3 directly interacts with GRP78. **a**, **b** Flag-GRP78 and Myc-OTUD3 were co-transfected into HEK293T cells. The cell lysates were subjected to immunoprecipitation (IP) with anti-Myc or anti-Flag antibodies. **c** GST pull-down assays were performed to indicate the direct interaction between OTUD3 and GRP78. **d** H1299 and A549 cell lysates were subjected to immunoprecipitation with control IgG or anti-GRP78 antibody. The immunoprecipitates were then detected using the indicated antibodies. **e**, **f** HEK293T cells transfected with the indicated Flag–DUBs were subjected to immunoprecipitation (IP) with anti-Flag antibodies. The lysates and immunoprecipitates were analyzed. **g** Schematic illustration of GRP78 structure. HEK293T cells transfected with the indicated constructs were subject to GST pull-down. The proteins retained on Sepharose were blotted with the GFP antibody. **h** Schematic illustration of OTUD3 structure. HEK293T cells transfected with GFP-GRP78 were subjected to GST pull-down. The proteins retained on Sepharose were blotted with the GFP antibody. For all panels, results are representative of three independent experiments. Uncropped images of blots are shown in Supplementary Fig. 9

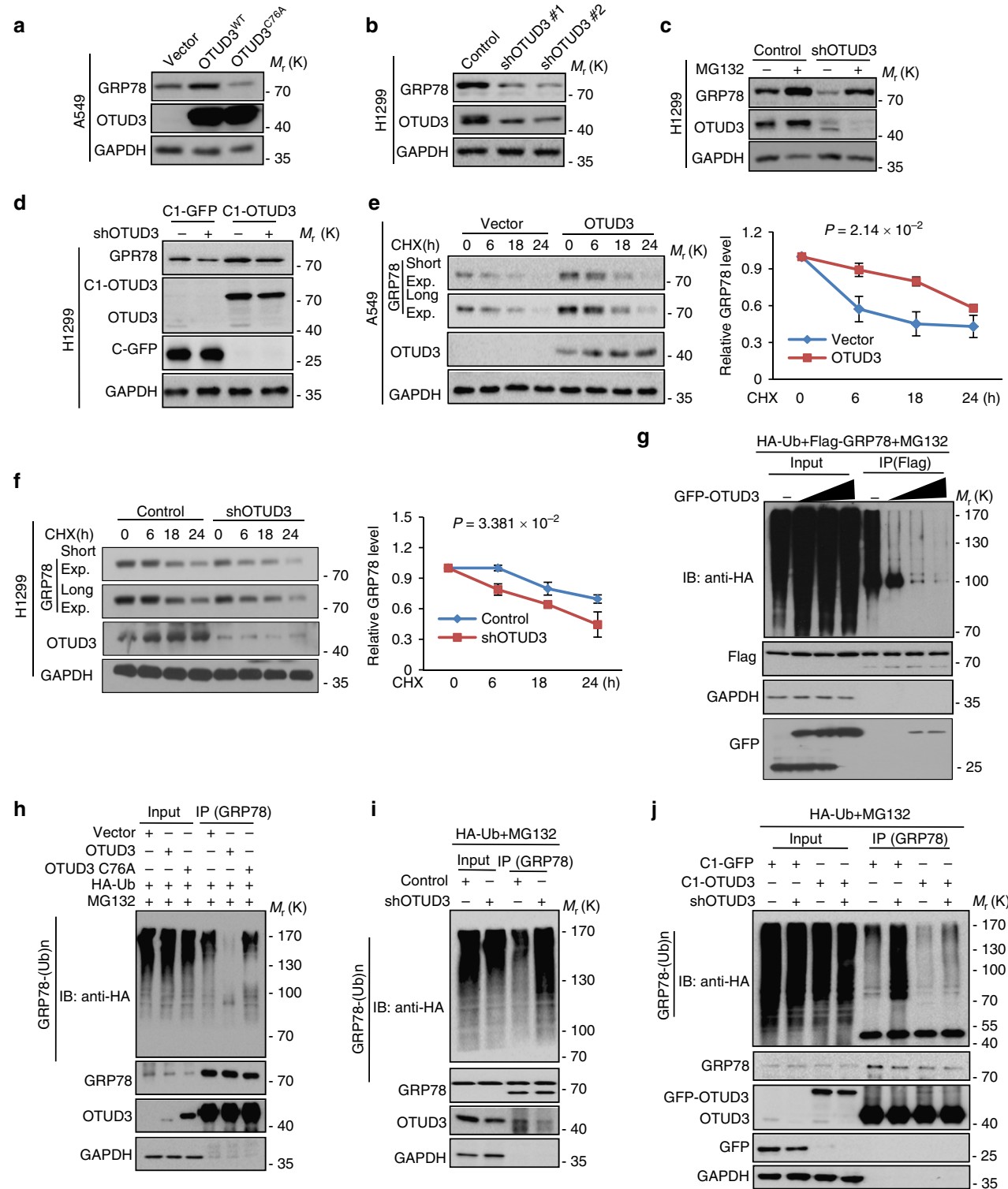

**Virus delivery into the lung**. The adeno-associated viruses expressing GFP (AAV-GFP) or Cre-recombinase-GFP (AAV-Cre-GFP) were purchased from Hanbio Co. LTD (Shanghai, China). Intratracheal delivery of adeno-associated virus was performed according to previously established protocol. In brief, 6- to 8-week-old animals were anesthetized using isoflurane and were set up in a biosafety cabinet. For intratracheal delivery, a gauge-24 catheter was inserted to the trachea, and virus solution in 75 μl sterile saline was pipetted to the top of the catheter to allow animal to gradually breathe in the solution. A titer of $1 \times 10^{11}$ viral genome copies was administered to each mouse. Animals after the procedure were kept warm using a heat lamp for recovery.

**Cell culture and reagents**. Cell lines H1299 and A549 (lung cancer); HEK293T (human embryonic kidney); HeLa (cervical cancer); HCT116 (colon cancer); HepG2 (human hepatocellular carcinoma (HCC)) and MCF7 (breast adenocarcinoma) were obtained from the American Type Culture Collection (ATCC), and authenticated by STR profiling and tested for mycoplasma contamination by GENEWIZ. A549, HeLa, MCF7, and HEK293T cells were cultured in DMEM (GIBCO-Invitrogen) supplemented with 10% FBS (fetal bovine serum). H1299 cells were cultured in 1640 (GIBCO-Invitrogen) supplemented with 10% FBS. Reagents used in this study included MG132 (Calbiochem), Puromycin (Mediatech), Cycloheximide (CHX) (Inalco Spa Mllano Italy), Polybrene (Sigma).

**Fig. 6** OTUD3 maintains GRP78 stability and de-polyubiquitylates GRP78. **a** OTUD3 WT and OTUD3 C76A were overexpressed into A549 cells. Cell lysates were subjected to detect with the indicated antibodies. **b** OTUD3 was depleted in the H1299 cells. Cell lysates were subjected to detect with the indicated antibodies. **c** H1299 cells with or without OTUD3 knockdown were treated with MG132 for 8 h. Cell lysates were subjected to analyze with the indicated antibodies. **d** C1-GFP or C1-OTUD3 were introduced into H1299 cells with or without endogenous OTUD3 knockdown. GRP78 levels were measured by western blotting. **e** A549 cells with or without overexpressing OTUD3 were treated with CHX (10 μg/ml) for the indicated times. The half-life of GRP78 were measured. **f** H1299 cells with or without depletion of OTUD3 were treated by CHX (10 μg/ml) for the indicated times. The half-life of GRP78 were measured. **g** 293 T cells were transfected with the indicated constructs were treated with MG132 for 8 h before collection. The whole-cell lysate was subjected to immunoprecipitation with Flag antibody and western blot with anti-HA antibody to detect ubiquitylated GRP78. **h–j** H1299 cells with or without overexpressing OTUD3 or OTUD3 C76A (**h**), with or without OTUD3 knockdown (**i, j**) were transfected with the indicated constructs and the cells then were treated with MG132 for 8 h before collection. The whole-cell lysate was subjected to immunoprecipitation with GRP78 antibody and western blot with anti-HA antibody to detect ubiquitylated GRP78. For all panels, data are representative results of three independent experiments. In **e**, **f**, quantification of GRP78 levels relative to GAPDH is shown. Resultsare shown as mean ± s.d; two-way ANOVA test. Statistics source data can be found in Supplementary Data 1. Uncropped images of blots are shown in Supplementary Fig. 9

**Plasmids and antibodies**. Full-length OTUD3 WT, full-length OTUD3 C76A, full-length GRP78, and full-length PTEN were cloned into the pCMV-Myc, pFlag-CMV-2 or pEGFP-C1 vectors as indicated. GST-tagged OTUD3, GST-tagged OTU + UBA (1–340 aa), UBA (184–340 aa), Tail (341–398 aa) truncations were cloned into the pGEX-4T-2 vector. G1 (1–124 aa), G2 (125–399 aa), G3 (281–499 aa), G4 (500–654 aa) truncations of GRP78 were cloned into the pEGFP-C1 vector. Antibodies used in immunoblotting were: anti-actin (1:1,000, sc-1616, Santa Cruz); GAPDH (1:1,000, sc-25778, Santa Cruz);anti-GRP78 (1:1000; 11587-1-AP, Proteintech); anti-OTUD3 (1:500; HPA028543, Sigma); anti-PTEN (1:1000; #9188, Cell Signaling) and anti-pSer473-AKT (1:1000; #4060, CellSignaling); Anti-Flag (1:1000; sc-965, Santa Cruz); anti-Myc (1:1000; sc-374171, Santa Cruz); anti-HA (1:1000; M180-3, MBL); anti-GFP (1:1000; 66002-1-Ig, Proteintech).Normal IgG (sc-2003, Santa Cruz).

**Lentivirus packaging and infection**. To generate the lentiviral shRNA constructs against human OTUD3, the OTUD3 and GRP78 shRNA sequences were cloned into the pLKO-puro vector. The shRNA sequences are reported in Supplementary Table 2. pLKO.1, pVSVG, pREV, and pGAG were co-transfected into HEK293T cells for 24 h, and cell culture media were collected. The full-length OTUD3 and C76A mutant sequences were cloned into the pCDH-puro vector. pCDH, pSPAX.2, and pMD.2 G were co-transfected into HEK293T cells for 24 h, and cell culture media were collected. The viruses were used to infect cells in the presence of polybrene. Forty-eight hours later, H1299 and A549 cells were cultured in medium containing puromycin for the selection of stable clones. The clones stably knocking down or overexpressing OTUD3 were identified and verified by western blotting.

**Cell transfections, immunoprecipitation, and immunoblotting**. Cells were transfected with indicated plasmids using Lipofectamine 2000 (Invitrogen) reagent according to the manufacturer's protocol. For immunoprecipitation assays, cells were lysed with NP40 lysis buffer (50 mM Tris-HCl, pH 8.0, 150 mM NaCl, 1% NP40, 0.5% deoxycholate) supplemented with protease-inhibitor cocktail (Biotool). Immunoprecipitations were performed using the indicated primary antibody and protein A/G agarose beads (Santa Cruz) at 4 °C. The immunocomplexes were then washed with 200 μl PBS for twice. Both lysates and immunoprecipitates were examined using the indicated primary antibodies followed by detection with the related secondary antibody and the Western Bright ECL chemiluminescent Detection Reagent (advansta). Uncropped scans of western blots are shown in Supplementary Fig. 9.

**GST pull-down assays**. Human cDNA for OTUD3 was cloned into a pGEX-4T-2 vector with an N-terminal GST-tag and purified from the *Escherichia coli* strain BL21 (Invitrogen) using GST Agarose beads. Bacterial-expressed GST, GST-OTUD3 or their truncates bound to glutathione-Sepharose 4B beads (from GE) was incubated with GFP-GRP78 expressed in HEK293T cells for overnight at 4 °C. Then the beads were washed with PBS four times, followed by western blotting.

**Protein half-life assay**. For the GRP78 half-life assay, the H1299 and A549 cells with stably expressing OTUD3 or the indicated shRNAs were treated with CHX (Sigma, 10 μg/ml) for the indicated durations before collection. Plasmids encoding OTUD3 WT were transfected into MCF7 and HeLa cells. Twenty-four hours later, the cells were treated with CHX for the indicated durations before collection.

**Fluorescence microscopy**. For detection of subcellular localization by immunofluorescence, after fixation with 4% paraformaldehyde and permeabilization in 0.2% Triton X-100 (PBS), cells were incubated with antibodies anti-OTUD3 (1:100; HPA028543, Sigma), anti-Calnexin (1:100, ab219644, abcam), anti-PTEN (1:50; 60300-1-lg, Proteintech), anti-GRP78 (1:100,YM1246, Immunoway) for 8 h at 4 °C, followed by incubation with secondary antibody (Invitrogen) for 1 h at room temperature. The nuclei were stained with DAPI (Sigma), and images were visualized with a Zeiss LSM 510 Meta inverted confocal microscope.

**In vivo GRP78 ubiquitylation assay**. For in vivo deubiquitylation assays, GFP-OTUD3, Flag-GRP78, and HA-ubiquitin were transfected into H1299 or HEK293T cells with Lipofectamine 2000. Twenty-four hours later, the cells were treated with 20 μM of the proteasome inhibitor MG132 (Calbiochem) for 8 h. Forty-eight hours later, cells were lysed with NP40 lysis buffer and incubated with anti-Flag antibody for 3 h and protein A/G agarose beads (Santa Cruz) for a further 6 h at 4 °C. Then the beads were washed three times with PBS buffer. The proteins were released from the beads by boiling in SDS-PAGE sample buffer and analysed by immunoblotting with anti-HA monoclonal antibody.

**RT-qPCR**. Total RNA was extracted with TRIzol (Invitrogen) and precipitated in ethanol. Total RNA (1 μg) was reverse transcribed into cDNA using SuperScript III First-Strand Synthesis SuperMix (Invitrogen, 11752-50). A 20 μl volume reaction consisted of 1 μl reverse transcription product and 250 nM of each primer. The primers used for the indicated gene products are described in Supplementary Table 1.

**Cell proliferation assay**. The HepG2, HCT116 and HeLa cells stably knocked down for OTUD3 were plated in 96-well plates (100 μl cell suspensions, $1 \times 10^4$ cells ml$^{-1}$). Cell numbers were estimated every 24 h by adding MTS (3-(4, 5-dimethylthiazol-2-yl)-5-(3-carboxy methoxyphenyl)-2-(4-sulphophenyl)-2H-tetrazolium, inner salt Sigma) to the wells 1 h before absorbance at 490 nm was measured. Each cell line was set up in four replicate wells, and the experiment was repeated three times.

**Colony formation assays**. H1299, A549, HepG2, HCT116, and HeLa cells with stable knockdown or overexpression of OTUD3 were harvested and pipetted well to become single-cell suspension in complete culture media in a given concentration (such as $1 \times 10^6$/ml). Dilute the single-cell suspension to 500 or 1000 cells in every well of 12-well plate. Put it at 37 °C with 5% $CO_2$ incubator for 2 weeks. Then the colonies were stained with 0.04% crystal violet-2% ethanol in PBS. Photographs of the stained colonies should be taken.

**Cell migration assay**. Cell migration assay was performed in 24-well transwell plate with 8-mm polyethylene terephalate membrane filters (Corning) separating the lower and upper culture chambers. In brief, H1299 or A549 cells were plated in the upper chamber at $5 \times 10^4$ cells per well in serum-free DMEM medium. The bottom chamber contained DMEM medium with 10% FBS. Cells were allowed to migrate for 24 h in a humidified chamber at 37 °C with 5% $CO_2$. After the incubation period, the filter was removed and non-migrant cells on the upper side of the filter were detached using a cotton swab. Filters were fixed with 4% formaldehyde for 15 min and cells located in the lower filter were stained with 0.1% crystal violet for 20 min and photographed.

**Tumor growth and metastasis assay**. BALB/c nude mice (6-week old, 18.0 ± 2.0 g) were obtained from Beijing Vital River Laboratory and were randomly divided into indicated groups. The mice in the groups were subcutaneously injected with the indicated cells stably expressing the indicated shRNAs or control. Tumor size was measured every 3 days by Vernier caliper and converted to TV according to the following formula: TV (mm$^3$) = (a × b$^2$)/2, where a and b are the largest and smallest diameters, respectively. All animals were killed 5 weeks after injection, and the transplanted tumors were removed, weighed, and fixed for further study. For metastasis assay, mice were injected through lateral tail vein.When the mouse was about to die, it was dissected to observe the metastatic organs.

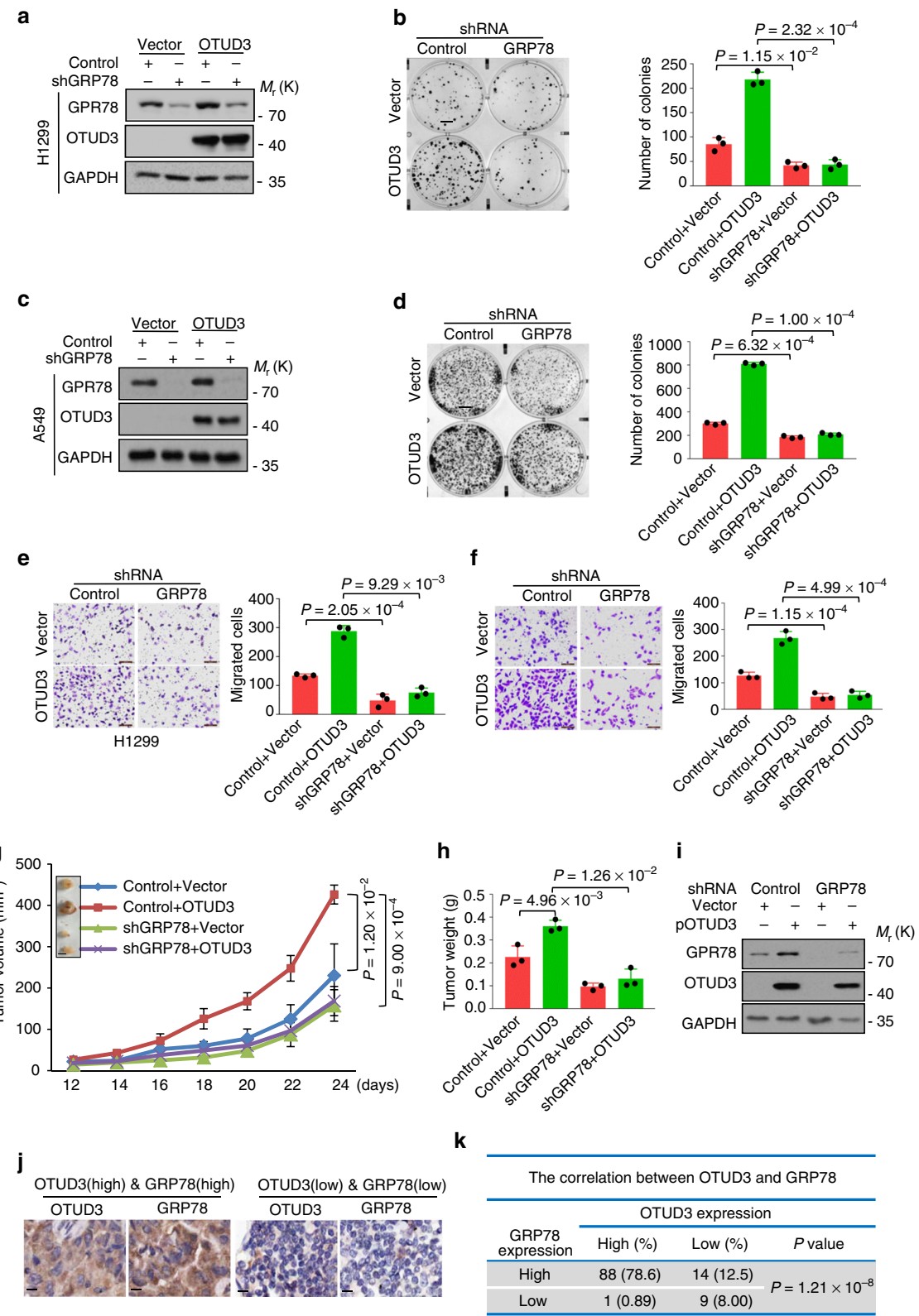

**Cohort and immunohistochemistry**. Tumor tissue microarrays, purchased from Shanghai Outdo Biotech Company, contain 73 pairs of hepatocellular carcinoma together with matched adjacent normal hepatocellular tissue, 73 pairs of colon cancer together with matched adjacent normal colon tissue, 30 pairs of cervical carcinoma together with matched adjacent normal cervical tissue, and 90 pairs of lung adenocarcinoma together with matched adjacent normal lung tissue and follow-up (range 0–120 months), respectively. Immunohistochemistry was performed by using the avidin-biotin complex method (Vector Labora-tories),

including heat-induced antigen-retrieval procedures. Incubation with antibodies against PTEN (1:150; #9188, Cell Signaling), OTUD3 (1:200; HPA028543, Sigma) was carried out at 4 °C for 12 h. All staining was assessed by a quantitative imaging method; the percentage of immunostaining and the staining intensity were recorded. An H-score was calculated using the following formula: H- $\text{SCORE} = \Sigma$ $(PI \times I) = $ (percentage of cells of weak intensity $\times$ 1) + (percentage of cells of moderate intensity $\times$ 2) + (percentage of cells of strong intensity $\times$ 3). PI indicates the percentage of positive cells vs all cells, and I represents the staining intensity.

**Fig. 7** OTUD3 promotes lung cancer cell growth and migration via GRP78. **a, b, e** GRP78 was knocked down in OTUD3-overexpressing H1299 cells. Cell lysates were analyzed by western blotting using the indicated antibodies (**a**). Cell growth and migration were examined by colony formation (scale bar, 1 cm) (**b**) and transwell assay (scale bar, 100 μm) (**e**). For **b, e**, results shown are mean ± s.d. $n = 3$ independent experiments. Data were analysed using Student's $t$-test. **c, d, f** GRP78 was knocked down in OTUD3-overexpressing A549 cells. Cell lysates were analyzed by western blotting using the indicated antibodies (**c**). Cell growth and migration were examined by colony formation (scale bar, 1 cm) (**d**) and transwell assay (scale bar, 100 μm) (**f**). For **d, f**, results shown are mean ± s.d. $n = 3$ independent experiments. Data were analysed using Student's $t$-test. **g, h** GRP78 was knocked down in H1299 cells with or without overexpressing OTUD3 and the cells then were subcutaneously injected ($1 \times 10^6$ cells per mouse) into 6-week-old nude mice ($n = 3$). Tumor formation was analysed. The data are shown as mean ± s.d. Statistical significance was calculated using two-way ANOVA test (**g**) and Student's $t$-test (**h**). (**g**, scale bar, 0.5 cm) **i** Immunoblotting of GRP78, OTUD3, and GAPDH in lysates of primary tumors from mice injected with the indicated H1299 cells. **j, k** Representative images from immunohistochemical staining of OTUD3 and GRP78 in lung cancer tissues ($n = 112$) and the correlation between OTUD3 and PTEN in human lung cancer was analyzed (**j**, scale bar, 20 μm). Statistics source data can be found in Supplementary Data 1. Uncropped images of blots are shown in Supplementary Fig. 9

**LC-MS/MS analysis**. OTUD3 proteins were immunoprecipitated with Protein A–G agarose from HEK293T cells. IP buffer (50 mM Tris–HCl pH 8.0, 100 mM NaCl, 1 mM EDTA, and 1% Nonidet P40) was used to precipitate Proteins in IP samples. A Q-Exactive HF MS (Thermo Fisher Scientific) interfaced with an Easy-nLC 1200 nanoflow LC system (Thermo Fisher Scientific) was used to analyze the samples from in-gel digestion. Ten microliter of loading buffer (5% methanol and 0.1% formic acid) was used to dissolve tryptic peptides. and 5 μl was loaded onto a homemade trap column (2 cm) packed with C18 reverse-phase resin at a maximum pressure of 280 bar with 12 μl of solvent A (0.1% formic acid in water). Peptides were separated on a silica microcolumn.

The MS analysis was performed in a data-dependent manner. The higher-energy collision dissociation (HCD) method was used to acquire the tandem mass spectra. Then raw files were searched against the human refseq protein database with further MS analysis. The mass tolerance of the precursor ions was set to 20 p.p.m and QE HF was set to 50 mmu. Protease digestion can not exceed two missed cleavages and the minimal required peptide length was seven amino acids. The data were also searched against a decoy database so that protein identifications were accepted at a FDR of 1%.

Protein identification data (accession numbers, peptides observed, sequence coverage) are available in Supplementary Data 2. All data and search results have been deposited to the iProX database (http://www.iprox.org)[44] with the iProX accession: IPX0001606000.

**Intrapulmonary implantation procedure**. Log-phase cell cultures of LLC cells were harvested with in phosphate buffered saline (PBS), washed three times with PBS, and resuspended at a cell density of $2 \times 10^6$/ml in PBS containing 500 μg/ml of Matrigel to prevent the suspension from leaking out of the lung. Animals were anaesthetized with narcotic (Ketamine 50 mg/ml and Xylazine 20 mg/ml). The left chest was swabbed with 75% alcohol and a small skin incision to the left chest wall (~5 mm in length) was made at about 5 mm tail side from the scapula. Subskin fat and muscles were separated from costal bones. On observing the left lung motion through the pleura, a 29-gauge needle attached to a 0.5 ml insulin syringe was directly inserted through the intercostal space into the lung to a depth of 3 mm. Tumor cells ($4 \times 10^4$) were injected into the lung parenchyma. The skin incision was closed with a surgical skin clip. They were returned to their cages.

**Statistical analysis**. The statistical significance of differences between various groups was calculated with the two-tailed paired $t$-test, and error bars represent standard deviation of the mean (s.d.). Statistical analyses, unless otherwise indicated, were performed using GraphPad Prism 5. Data are shown as mean ± s.d.

## Data availability

Source data for Figs. 1b, d, f, h, 2b, e, h, k, 3a, b, e, 4a-i, 6e, f, 7b-h and Supplementary Figs. 1j-l, 2c-f, i, j, 4a-d, 7a–c and 8c, g have been provided as Supplementary Data 1. All IP-MS data and search results have been deposited to the iProX database (https://www.iprox.org/page/project.html?id=IPX0001606000) with the iProX accession: IPX0001606000. All data are included within the article, supplementary information or available from the authors upon reasonable request.

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

## Acknowledgements

This work was supported by National Key R&D Program of China (2017YFA0505602), Chinese National Natural Science Foundation Projects (31330021, 81702743, 81874136, and 81402260), the National Key Technologies R&D Program for New Drugs (2015ZX09J15102), and Beijing Natural Science Foundation Project (Z151100003915083).

## Author contributions

L.Z. and C.H. conceived the project and performed the project planning. C.H. and T.D. performed deubiquitylation, OTUD3-GRP78 interaction and molecular mechanism investigations. H.L. and Y.F. performed the main experimental work of mouse models and IHC analysis. C.H. and T.D. participated in OTUD3 tumor function study in lung cancer. L.Y. and H.L. participated in constructing OTUD3 knockdown hepatocellular cancer, colon cancer and cervical cancer cell lines and the related cell proliferation, colony formation, and xenograft assays. X.G., Q.Z., and Y.Y. participated in plasmid construction. X.Y. and Z.L. contributed to the NanoSPECT images analysis. X.L and C.L contributed to the Micro-CT images analysis. C.P.C. contributed to the IHC analysis and data analysis. L.Z., C.H., T.D. and H.L. designed the experiments, analyzed the data, and wrote the manuscript.

## Additional information

**Competing interests:** The authors declare no competing interests.

