## [Peer Review File · Nature Communications]

Reviewers' comments:

Reviewer #1, Expertise: DUBs (Remarks to the Author):

The manuscript by Han et al. describes novel function of deubiquitinase OTUD3 in promoting lung cancer growth and progression and provides basic mechanistical insights. Based on their previous work indentifying OTUD3 as a tumor suppressor in the breast cancer the authors extended the investigation to additional types of cancers. At least in colon, liver and cervical cancers OTUD3 seems to preserve its tumor suppressive function. Surprisingly, depletion of OTUD3 led to inhibition of lung adenocarcinoma formation in inducible KRASG12D mice. Accordingly, examination of human tissues indicated on a positive role of OTUD3 in promoting lung adenocarcinoma. Mechanistically, They suggest that OTUD3 stabilises the tumorigenic glucose-regulated protein 78 (GRP78). The authors used multiple techniques including in vivo mice models, examination of human tissues sections, in vitro analysis of stable human cell lines, and several biochemical experiments to support their conclusions. The findings are novel and will be of a high interest to researchers from both cancer research and ubiquitin fields. The work also provides very interesting and novel insigths into tissue-dependent functions of OTUD3 with a potential future clinical relevance.

Overall, the manuscript is well written and most of experiments and data analysis are performed accurately. Though, I have several remarks that should be addressed before publishing the manuscript.

Major comments:

1. Authors claim that OTUD3 is a potent deubiquitinase for PTEN. However, except of WAT and muscle tissues PTEN protein levels are not changed. In contrary, downregulation of OTUD3 in multiple cancer cell lines leads to drop in PTEN levels. This suggests that under normal physiological conditions OTUD3 does not affect PTEN stability and the PTEN protection activity of OTUD3 is likely initiated only by already established cancerous signaling. Authors do not exclude this possibility but I feel this phenomenon should be more highlighted in the text, at least in the discussion.
2. One of the major finding of the manuscript is the identification of GRP78 as a novel target of OTUD3. The experimental evidence pointing on the function relationship between these two proteins is quite strong. However, the authors do not provide any clear evidence that OTUD3 directly interacts with GRP78. Experiments using recombinant GST-tagged OTUD3 (Fig 5c,g,h) use whole cell lysate with overexpressed Flag or GFP-tagged GRP78. To definitely prove direct interaction between OTUD3 and GRP78, for example both proteins could be purified from bacterial system and an in vitro pull down assay could be performed. Also, the Fig. 5c is of a low quality (contaminating Flag-GRP78 band in the control GST sample) and should be removed or ideally changed as suggested above.
3. The statement that OTUD3 is the only deubiquitinase from the OTU family is not fully backed up by experimental evidence. For example, the two most closely related deubiquitinases (OTUD1 and OTUD2) are missing in the co-immunoprecipitation experiments (Fig. 5d and Supp Fig. 5d). Additionally, OTUD4 and OTUD5 are also neglected from the list.
4. Multiple experiments indicate that GRP78 is downregulated in lung adenocarcinoma cells expressing shOTUD3. However, the GRP78 levels are identical in control and shOTUD3 cells in the Fig. 6f at the 0hrs of cycloheximide treatment. Could the authors explain this discrepancy?

Minor comments:

1. The microscopy Fig. 5e is of a low quality and the colocalisation is not clearly visible. Please

provide at least a close up view and highlight the sites of colocalisation.

2. Why a part of GFP immunoblot is missing in the IP fraction?

3. typing mistakes:

Generally, when referring to cancerous tissues the authors use both terms „tumor“ and „tumour“ in the main text and in the Figure labels. It would be good to stay consistent and use only a single version throughout the manuscript.

page 4, line 62: stabiilizes (should be stabilizes)

page 7, line 148: Tumor (should be tumor)

page 10, line 213: Supplementary Fig. 2d,e (should be Supplementary Fig. 3d,e)

page 11, line 221: deubiquitylase (should be deubiquitylase)

page 12, line 238: parteners (should be partners)

page 12, line 246: protein (should be proteins)

Figure 4 f – missing labell (f)

Reviewer #2, Expertise: Cancer (lung, breast), mouse models (Remarks to the Author):

Han and colleagues analyzed the role of OTUD3 on lung and breast tumors. In agreement with their previously published data deletion of OTUD3 accelerates breast tumorigenesis in the MMTV-PyMT mouse model. In contrast, deletion of OTUD3 suppressed lung adenocarcinoma formation in the KRASG12D model and BAC-mediated overexpression of OTUD3 accelerated tumorigenesis in this model. Supporting this findings, human lung tumors show higher OTUD3 expression levels compared to non-tumor tissue and high expression of OTUD3 correlated with bad prognosis. Using lung cancer cell lines, the authors show that downregulation of OTUD3 impairs cell proliferation and tumorigenesis. Mechanistically, the authors show that OTUD3 interacts and stabilize GPR78, which has been identified as a pro-tumorigenic gene candidate in several cancers. Consequently, knock-down of GPR78 prevented the pro-tumorigenic effect of OTUD3.

Major concerns

1) The authors should confirm that PTEN levels are downregulated in mammary gland tumors lacking OTUD3 derived from the MMTV-PyMT mouse models

2) Figure 1 i-j-k. Show a quantification of tumor number, tumor area and tumor grade in the KRASG12D wt and OTUD3 deficient mouse models

3) Figure 1 l-m-n. OTUD3 overexpressing mice on the KRASG12D background: show a Kaplan Meier experiment, as well as a quantification of tumor number, tumor area and tumor grade.

4) Figure 4 f. Xenograft models of H1299 and A549 cells. Tumors derived from H1299 control shRNA grow up to 2000 mm³ in 24 days. Tumors derived from H1299 vector control grow up to more or less 50 mm³ in 24 days. Both controls should behave more or less similar. The differences are too big and indicate that something is not controlled in this experiment. Same issue on the A549 cells. Tumor size in the controls of the knockdown and overexpression experiments differ enormously. Please, address this issue.

5) Figure 5 e. GRP78 is a protein localized mainly in the ER. An ER marker should be included in the co-localization experiments and high magnification and resolution figures should be provided.

6) I find it intriguing that OTUD3 stabilizes PTEN and suppresses tumorigenesis in breast and other tumors but not in lung cancer. Conversely, OTUD3 stabilize GRP78 in lung tumors but no other

tumors. Is there different adaptor proteins that mediates the interaction between OTUD3 and PTEN/GRP78 in different tumors? On this line, PTEN is mainly localized in the cytoplasm but barely in the ER, while GRP78 is mainly located in the ER. Is the sub-cellular localization of OTUD3 different in different cancer cell lines? In other words, is OTUD3 interacting with PTEN in the cytoplasm (non ER) of breast cancer cells while in lung cancer cells OTUD3 interacts with GRP78 in the ER? This should be addressed by performing co-localization experiments in lung and breast cancer cell lines using ER markers as mentioned in point 5

7) The authors used a mouse model that lack OTUD3 in all cell types (a "global" knock-out) or overexpress OTUD3 in all cell types (BAC transgenic). Although the authors demonstrated a tumor cell intrinsic function of OTUD3, they cannot rule out that lack or overexpression of OTUD3 in non-tumor cells (tumor stroma) influence tumorigenesis in the in vivo KRASG12 mouse model. To assess the impact of OTUD3 in non-tumor cells, the authors should orthotopically transplant a mouse lung cancer cell line (e.g. 368T1 doi: 10.1101/gad.222745.113) in control mice and mice lacking and overexpressing OTUD3 and assess tumor development.

Minor points

1) Sometimes figure numbers are mislabeled. Please go thought and correct it.

2) Statistics: In some of the figures, ANOVA test is used to compare two groups while Student t-test is used to compare three groups. It should be the other way around.

Response letter

**Re: Deubiquitylase OTUD3 stabilizes GRP78 and promotes lung tumorigenesis
(Manuscript ID NCOMMS-18-27383-T)**

Reviewer #1 (Remarks to the Author):

The manuscript by Han et al. describes novel function of deubiquitinase OTUD3 in promoting lung cancer growth and progression and provides basic mechanistical insights. Based on their previous work indentifying OTUD3 as a tumor suppressor in the breast cancer the authors extended the investigation to additional types of cancers. At least in colon, liver and cervical cancers OTUD3 seems to preserve its tumor suppressive function. Surprisingly, depletion of OTUD3 led to inhibition of lung adenocarcinoma formation in inducible KRASG12D mice. Accordingly, examination of human tissues indicated on a positive role of OTUD3 in promoting lung adenocarcinoma. Mechanistically, They suggest that OTUD3 stabilises the tumorigenic glucose-regulated protein 78 (GRP78). The authors used multiple techniques including in vivo mice models, examination of human tissues sections, in vitro analysis of stable human cell lines, and several biochemical experiments to support their conclusions. The findings are novel and will be of a high interest to researchers from both cancer research and ubiquitin fields. The work also provides very interesting and novel insigths into tissue-dependent functions of OTUD3 with a potential future clinical relevance.

Overall, the manuscript is well written and most of experiments and data analysis are performed accurately. Though, I have several remarks that should be addressed before publishing the manuscript.

Response: We thank the reviewer for the kind comments on the thoroughness of our manuscript and for recognizing the novelty and significance of this study. According

to the concerns, we have revised the manuscript and the responses are listed below point by point.

(Major Points)

1. Authors claim that OTUD3 is a potent deubiquitinase for PTEN. However, except of WAT and muscle tissues PTEN protein levels are not changed. In contrary, downregulation of OTUD3 in multiple cancer cell lines leads to drop in PTEN levels. This suggests that under normal physiological conditions OTUD3 does not affect PTEN stability and the PTEN protection activity of OTUD3 is likely initiated only by already established cancerous signaling. Authors do not exclude this possibility but I feel this phenomenon should be more highlighted in the text, at least in the discussion.

Response: We sincerely thank the reviewer for such insightful comments. They are important points. In the revised manuscript, we added the description. Actually, we have compared the effect of OTUD3 on PTEN expression in different cancer cells in Supplementary Fig. 3. Depletion of OTUD3 dramatically decreased PTEN levels, and two independent OTUD3 shRNAs showed significant effects (Supplementary Fig. 3a,b). Moreover, ectopic expression of OTUD3 resulted in PTEN protein elevation in a dose-dependent manner in these cells (Supplementary Fig. 3c). And we have also detected the expression of p-AKT(S473), AKT and PTEN in tissues from OTUD3 WT and KO littermates (2 months) in Supplementary Fig. 1f. However, except of WAT and muscle tissues PTEN protein levels are not changed in other examined tissues. After that we also analyzed the expression of p-AKT(S473), AKT and PTEN in breast cancer tissues from PyMT/OTUD3 WT and PyMT/OTUD3 KO mice and found that PTEN protein levels were significantly decreased and p-AKT (Ser-473) levels were increased. These data were shown in Fig.1g and Supplementary Fig. 1g.

Thus, we absolutely agree that the stabilization of PTEN by OTUD3 might only be initiated in those already established cancerous tissues or cells as the reviewer mentioned. Moreover, under physiological condition, we hypothesize that deletion of OTUD3 or USP13 alone in mice would not result in a significant down-regulation of the PTEN protein level since they synergistically stabilize PTEN. Generation of the OTUD3/USP13 double knockout mice would be a better way to investigate the

physiological role of OTUD3 and USP13 in regulating PTEN. These discussions have been added into the manuscript.

2. One of the major finding of the manuscript is the identification of GRP78 as a novel target of OTUD3. The experimental evidence pointing on the function relationship between these two proteins is quite strong. However, the authors do not provide any clear evidence that OTUD3 directly interacts with GRP78. Experiments using recombinant GST-tagged OTUD3 (Fig 5c,g,h) use whole cell lysate with overexpressed Flag or GFP-tagged GRP78. To definitely prove direct interaction between OTUD3 and GRP78, for example both proteins could be purified from bacterial system and an in vitro pull down assay could be performed. Also, the Fig. 5c is of a low quality (contaminating Flag-GRP78 band in the control GST sample) and should be removed or ideally changed as suggested above.

Response: We thank the reviewer for bringing up this constructive suggestion. To definitely prove the direct interaction between OTUD3 and GRP78 in vitro, we first purified GST-OTUD3 and His-GRP78 from bacterial system and then an in vitro pull-down assay was performed. The result indicated that OTUD3 directly interacted with GRP78 under cell-free condition (Fig.5c). And the Fig. 5c which is of a low quality was changed.

In addition, we also purified GST-OTUD3-OTU, GST-OTUD3-UBA-Tail, His-GRP78-G123 and His-GRP78-G4 truncations from bacterial system to perform mapping assay in vitro. These results also revealed that the C-terminal region (aa 500-654) of GRP78 was critical for interaction with OTUD3 (Supplementary Fig 5e). Similarly, the N-terminal OTU region of OTUD3 mediated the physical interaction with GRP78 (Supplementary Fig 5f).

3. The statement that OTUD3 is the only deubiquitinase from the OTU family is not fully backed up by experimental evidence. For example, the two most closely related deubiquitinases (OTUD1 and OTUD2) are missing in the co-immunoprecipitation experiments (Fig. 5d and Supp Fig. 5d). Additionally, OTUD4 and OTUD5 are also neglected from the list.

Response: We thank the reviewer for raising this outstanding concern. To fully prove that whether OTUD3 specifically interacted with GRP78, we carried out the co-immunoprecipitation experiments. Among the examined nine OTUs (i.e. OTUD1, OTUD2, OTUD3, OTUD4, OTUD5, OTUD6A, OTUD6B, OTUD7B and OTULIN), only OTUD3 was able to bind to GRP78 (Fig. 5e,f).

4. Multiple experiments indicate that GRP78 is downregulated in lung adenocarcinoma cells expressing shOTUD3. However, the GRP78 levels are identical in control and shOTUD3 cells in the Fig. 6f at the 0hrs of cycloheximide treatment. Could the authors explain this discrepancy?

Response: Thank you! This issue should come from the long exposure time during the immunoblotting analysis. When the exposure is strong, it's hard to identify the difference of the expression of GRP78 between the control and shOTUD3 cells at 0 hour timepoint. We have repeated the experiment, optimized the exposure time and provided new data in the revised Fig. 6f.

(Minor pints)

1. The microscopy Fig. 5e is of a low quality and the colocalisation is not clearly visible. Please provide at least a close up view and highlight the sites of colocalisation.

Response: According to your suggestion, we analyzed the localization of OTUD3 and GRP78 using immunofluorescence and used Calnexin as a ER marker. We observed their co-localization in cytoplasm especially in ER region (Supplementary Fig. 6l).

2. Why a part of GFP immunoblot is missing in the IP fraction?

Response: We thank the reviewer for the important suggestion. In the revised manuscript, we added the missing parts of GFP immunoblot in Fig. 6g,j.

3. typing mistakes:

Generally, when referring to cancerous tissues the authors use both terms „tumor“ and „tumour“ in the main text and in the Figure labels. It would be good to stay consistent and use only a single version throughout the manuscript.

page 4, line 62: stabiiilizes (should be stabilizes)

page 7, line 148: Tumor (should be tumor)

page 10, line 213: Supplementary Fig. 2d,e (should be Supplementary Fig. 3d,e)

page 11, line 221: deubiquitylase (should be deubiquitylase)

page 12, line 238: parteners (should be partners)

page 12, line 246: protein (should be proteins)

Figure 4 f – missing labell (f)

Response: We thank the reviewer for raising these comments. We checked and corrected the errors in the manuscript. Once again, thank you very much for all of the comments and questions.

Reviewer #2 (Remarks to the Author):

Han and colleagues analyzed the role of OTUD3 on lung and breast tumors. In agreement with their previously published data deletion of OTUD3 accelerates breast tumorigenesis in the MMTV-PyMT mouse model. In contrast, deletion of OTUD3 suppressed lung adenocarcinoma formation in the KRASG12D model and BAC-mediated overexpression of OTUD3 accelerated tumorigenesis in this model. Supporting this findings, human lung tumors show higher OTUD3 expression levels compared to non-tumor tissue and high expression of OTUD3 correlated with bad prognosis. Using lung cancer cell lines, the authors show that downregulation of OTUD3 impairs cell proliferation and tumorigenesis. Mechanistically, the authors show that OTUD3 interacts and stabilize GPR78, which has been identified as a pro-tumorigenic gene candidate in several cancers. Consequently, knock-down of GPR78 prevented the pro-tumorigenic effect of OTUD3.

Response: We thank the reviewer for the kind comments on the thoroughness of our manuscript and for recognizing the novelty of this study.

(Major Points)

1. The authors should confirm that PTEN levels are downregulated in mammary gland tumors lacking OTUD3 derived from the MMTV-PyMT mouse models.

Response: We fully agree with the reviewer's advice and thank the reviewer for raising the insightful concern. The expression levels of PTEN and p-AKT (Ser-473) were detected in breast cancer tissues from PyMT/WT and PyMT/OTUD3 KO mice by WB and IHC. As shown in Fig.1g and Supplementary Fig. 1g, decreased PTEN protein level and increased p-AKT (Ser-473) level were observed in PyMT/OTUD3 KO mice, compared with PyMT/OTUD3 WT mice tissues.

2. Figure 1 i-j-k. Show a quantification of tumor number, tumor area and tumor grade in the KRAS^{G12D} wt and OTUD3 deficient mouse models.

Response: We thank the reviewer for raising the insightful concern. We have added a quantification of tumor number, tumor area and tumor grade in the KRAS^{G12D} / OTUD3 WT and OTUD3 KO mouse models in Supplementary Fig. 1j-l.

3. Figure 1 l-m-n. OTUD3 overexpressing mice on the KRAS^{G12D} background: show a Kaplan Meier experiment, as well as a quantification of tumor number, tumor area and tumor grade.

Response: We thank the reviewer for raising the insightful concern. We have added a quantification of tumor number, tumor area and tumor grade in OTUD3 overexpressing mice on the KRAS^{G12D} background and the Kaplan Meier experiment in Supplementary Fig. 2c-f.

4. Figure 4 f. Xenograft models of H1299 and A549 cells. Tumors derived from H1299 control shRNA grow up to 2000 mm³ in 24 days. Tumors derived from H1299 vector control grow up to more or less 50 mm³ in 24 days. Both controls

should behave more or less similar. The differences are too big and indicate that something is not controlled in this experiment. Same issue on the A549 cells. Tumor size in the controls of the knockdown and overexpression experiments differ enormously. Please, address this issue.

Response: We thank the reviewer for raising the insightful concern. We used different amounts of cell intentionally in the knockdown group and the overexpression group for xenograft models of H1299 and A549. In the knockdown group, 2×10^6 H1299 or A549 cells per mouse were used and in the overexpression group, 1×10^6 H1299 or A549 cells per mouse were used. We have added the information in the Figure legends.

5. Figure 5 e. GRP78 is a protein localized mainly in the ER. An ER marker should be included in the co-localization experiments and high magnification and resolution figures should be provided.

Response: Thank you very much. According to your suggestion, we analyzed the localization of OTUD3 and GRP78 using immunofluorescence and used Calnexin as a ER marker. We observed their co-localization in cytoplasm especially in the ER region in H1299 cells (Supplementary Fig. 6l).

6. I find it intriguing that OTUD3 stabilizes PTEN and suppresses tumorigenesis in breast and other tumors but not in lung cancer. Conversely, OTUD3 stabilize GRP78 in lung tumors but no other tumors. Is there different adaptor proteins that mediates the interaction between OTUD3 and PTEN/GRP78 in different tumors? On this line, PTEN is mainly localized in the cytoplasm but barely in the ER, while GRP78 is mainly located in the ER. Is the sub-cellular localization of OTUD3 different in different cancer cell lines? In other words, is OTUD3 interacting with PTEN in the cytoplasm (non ER) of breast cancer cells while in lung cancer cells OTUD3 interacts with GRP78 in the ER? This should be addressed by performing co-localization experiments in lung and breast cancer cell lines using ER markers as mentioned in point 5

Response: We fully agree with the reviewer's advice and thank the reviewer for raising the insightful concern. We analyzed the localization of OTUD3 and GRP78/PTEN using immunofluorescence in breast cancer cell and lung cancer cell and used Calnexin as a ER marker (Supplementary Fig. 6l-n). We found that OTUD3 was co-localized with GRP78 especially in the ER in lung cancer cell (Supplementary Fig. 6l). OTUD3 and PTEN did not show the ER subcellular co-localization in lung cancer cell (Supplementary Fig. 6m). On the contrary, OTUD3 did not co-localize with GRP78 in breast cancer cell (Supplementary Fig. 6n). Collectively, these data suggest that GRP78 specifically co-localized with OTUD3 in the ER of lung cancer cells.

7. The authors used a mouse model that lack OTUD3 in all cell types (a “global” knock-out) or overexpress OTUD3 in all cell types (BAC transgenic). Although the authors demonstrated a tumor cell intrinsic function of OTUD3, they cannot rule out that lack or overexpression of OTUD3 in non-tumor cells (tumor stroma) influence tumorigenesis in the in vivo KRASG12 mouse model. To assess the impact of OTUD3 in non-tumor cells, the authors should orthotopically transplant a mouse lung cancer cell line (e.g. 368T1 doi: 10.1101/gad.222745.113) in control mice and mice lacking and overexpressing OTUD3 and assess tumor development.

Response: We thank the reviewer for raising the insightful concern. To evaluate the impact of OTUD3 in non-tumor cells (tumor stroma), we orthotopically transplanted Lewis lung carcinoma cells (LLC) in OTUD3 WT, KO and TG mice. Firstly, LLC cells were injected into the left lung of OTUD3 WT and KO mice in situ, respectively. Then we imaged a separate cohort of OTUD3 WT and OTUD3 KO mice at 5, 10 and 15 days using Micro-CT (Micro-Computed Tomography) imaging. OTUD3 KO mice showed less detectable lesions than OTUD3 WT mice in the lung by coronal and transverse section imaging μ CT scan (Supplementary Fig. 8a). We also performed tumor analysis in mice at 10 and 15 days. OTUD3 KO mice exhibited a dramatically decreased tumor burden compared with that of the OTUD3 WT mice (Supplementary Fig. 8b,d). The weight of lung and tumor tissues from OTUD3 KO mice were significantly smaller than those of OTUD3 WT mice at 15 days (Supplementary Fig. 8c). To further confirm the carcinogenic function of OTUD3 in non-tumor cells, we

orthotopically transplanted LLC cells in OTUD3 WT and TG mice. Following analysis demonstrated that OTUD3 TG mice displayed more detectable lesions than OTUD3 WT mice in the lung (Supplementary Fig. 8e-h). These data suggest that OTUD3 not only plays a oncogenic role in lung cancer cells, but also promotes the development of lung cancer by affecting the growth environment of lung cancer cells.

Minor points

1) Sometimes figure numbers are mislabeled. Please go thought and correct it.

Response: Thanks for your suggestion, we have corrected it in Figure 4 and carefully checked all the figures labels.

2) Statistics: In some of the figures, ANOVA test is used to compare two groups while Student t-test is used to compare three groups. It should be the other way around.

Response: Thank you for pointing out this issue. We carefully checked all the statistics and ensured they were correctly used. Once again, thank you very much for all of the comments and questions.

REVIEWERS' COMMENTS:

Reviewer #1 (Remarks to the Author):

The authors answered all my questions and significantly improved the manuscript. I have no additional comments and support publishing of the manuscript in Nature Communications.

Reviewer #2 (Remarks to the Author):

The authors have addressed all my concerns and the manuscript has improved. I am happy to recommend its publication.

REVIEWERS' COMMENTS:

Reviewer #1 (Remarks to the Author):

The authors answered all my questions and significantly improved the manuscript. I have no additional comments and support publishing of the manuscript in Nature Communications.

Response: We sincerely thank the reviewer for the kind comments and the recommendation for publication.

Reviewer #2 (Remarks to the Author):

The authors have addressed all my concerns and the manuscript has improved. I am happy to recommend its publication.

Response: Once again, we thank the reviewer for the positive comments of our revised manuscript and our efforts in improving the integrity of the data.